
# High-Resolution Data Assimilation for Two Maritime Extreme Weather
# Events: A comparison between 3DVar and EnKF.
Diego S. Carrió[1], Vincenzo Mazzarella[2], Rossella Ferretti[2]
*[1]Meteorology Group, Department of Physics, University of the Balearic Islands, Palma, Spain*
*[2]CETEMPS, Department of Physical and Chemical Sciences, University of L'Aquila, L'aquila 67100,*
*Italy*
## Abstract
*Populated coastal regions in the Mediterranean are known to be severely affected by extreme weather*
*events. Generally, they are initiated over maritime regions, where a lack of in-situ observations is*
*present, hampering the initial conditions estimations and hence, the forecast accuracy. To face this*
*problem, Data Assimilation (DA) is used to improve the estimation of the initial conditions and their*
*respective forecasts. Although comparisons between different DA methods have been performed at*
*global scales, few studies are performed at high-resolution, focusing on extreme weather events*
*triggered over the sea and enhanced by complex topographic regions. In this study, we investigate the*
*role of assimilating different types of conventional and remote-sensing observations using the*
*variational 3DVar and the ensemble-based EnKF, which are of the most common DA schemes used*
*globally at National Weather Centers. To this aim, two different events are chosen because of both the*
*different areas of occurrence and the triggering mechanisms. Both the 3DVar and the EnKF are used*
*at convection permitting scales to improve the predictability of these two high-impact coastal extreme*
*weather episodes, which were poorly predicted by numerical weather prediction models: (a) the heavy*
*precipitation event IOP13 and (b) the intense Mediterranean Tropical-like cyclone Qendresa. Results*
*show that the EnKF and 3DVar perform similarly for the IOP13 event for most of the verification*
*metrics, although looking at the ROC and AUC scores, the EnKF clearly outperforms the 3DVar.*
*However, the ensemble mean of the EnKF is in general worse than the 3DVar for Qendresa, although*
*some of the ensemble members of the EnKF individually outperforms the 3DVar allowing for gaining*
*information on the physics of the event and hence the benefits of using an ensemble-based DA scheme.*
*Correspondence:* Diego S. Carrió, University of the Balearic Islands, 07122, Cra. Valldemossa km 7.5,
Balearic Islands, Palma (Spain)
*Email:* diego.carrio@uib.es
*Keywords*: Variational Data Assimilation (3DVar), ensemble data assimilation (EnKF), low-predictable
weather events, extreme weather events, high-resolution numerical forecasts.



## 1. Introduction

The Mediterranean basin is recognized as one of the geographical regions most frequently affected by high impact weather events in the world (Petterssen, 1956). The Mediterranean region has a natural disposition for these events because of its singular orographic features, which include having a relatively warm sea surrounded by complex terrain. This geographical configuration forces the warm and moist airflow to lift, favoring condensation and triggering convection. Hazardous weather events in this region, such as heavy precipitation (e.g., flash floods, snowstorms), cyclogenesis or windstorms (e.g., squall lines, tornadic thunderstorms), produce huge economic, injury and human losses in populated coastal regions (e.g., Romero et al., 1998b; Llasat and Sempere-Torres, 2001; Llasat et al., 2010; Jansa et al., 2014; Flaounas et al., 2016; Pakalidou and Karacosta, 2018; Amengual et al., 2021). Since 1900, more than 500 billion Euros associated with total damages to the property and over 1.3 million fatalities related to hydrometeorological disasters has been registered for the EM-DAT international disaster database[1]. These effects underscore the critical need for accurate and rapid high-resolution weather forecasting systems, aimed at extending the lead time for severe weather warnings, thereby enabling the implementation of effective mitigation strategies to reduce fatalities and economic losses. However, while the accuracy of weather forecasting has significantly improved in recent years, with better representation of physical processes and dynamics, accurate prediction of high impact weather events in terms of their location, timing, and intensity remains a major challenge for the scientific community (Stensrud et al., 2009; Mass et al., 2002; Bryan and Rotunno, 2005; Yano et al., 2018; Torcasio et al., 2021). For this reason, improving the forecast of high-impact weather events becomes an imperative goal.

Deficiencies in the accurate prediction of the location (spatial and temporal), intensity and phenomenology of extreme weather events are tightly related to the accuracy of the initial conditions of the system (Wu et al., 2013). The initial conditions of the hazardous weather events affecting coastal populated regions, are typically poorly estimated, mainly because these weather systems originate over the sea, where there is a lack of *in-situ* observations. Enhanced representations of the initial conditions are typically achieved by blending information from observations into numerical models through sophisticated *Data Assimilation* (DA) techniques (Kalnay, 2003), which accounts not only for the nominal values of the observations and the model, but also accounts for their respective error statistics. DA has been widely used and applied for global numerical weather prediction (NWP) problems (e.g., Eliasen, 1954; Lorenc, 1981; Le Dimet and Talagrand, 1986; Rabier et al., 2000, Whitaker et al., 2008, Carrassi et al., 2018; Albergel, et al., 2020, among others). However, less attention has been paid to convective-scale NWP problems, especially those associated with small scale convective phenomena initiated over regions with sparse observational data coverage, such as the extreme weather events affecting coastal regions in the Mediterranean basin (Carrió et al., 2016; Amengual et al., 2017; Mazzarella et al., 2021). To improve forecasts of such extreme weather events, accurate high resolution numerical weather models which solve convective scale processes are required, as well as dense observations at high spatial and temporal resolution. These will provide accurate information regarding the convective systems themselves or their environmental conditions. One of the most important sources of convective scale information are ground weather radars that provide three-dimensional data related to the storms at high spatial (order of hundreds of meters) and temporal (order of few minutes) resolution. In addition, weather radars provide thermodynamic and dynamic information of thunderstorms, which are crucial to understand and forecast convective structures. Due to the high spatio-temporal variability of convective structures, a rapid update cycle of the initial state (i.e.,

---

[1] https://www.emdat.be/



analysis) using weather radar observations is required to reduce errors and keep physical
balances in the initial conditions. Several studies have shown the positive impact in forecasting
severe weather events by assimilating weather radar information (e.g., Xiao and Sun, 2007;
Lee et al., 2010; Wheatley et al., 2012; Yussouf et al., 2015; Carrió et al., 2019; Mazzarella et
al., 2021).
During the last decades, different DA algorithms have been developed with the aim of
improving weather forecasts making use of all available observations in the best possible way.
In this context, most of the developed DA methods are based on exploiting Bayes' Theorem
(Lorenc, 1986) and making use of different types of approximations. Generally, DA algorithms
can be classified into the following three Bayesian-based families: (a) Variational DA (e.g.,
3DVar (Barker et al., 2004) or 4DVar (Huang et al., 2009)); (b) Ensemble-based DA, which
are based on the *Ensemble Kalman Filter* (EnKF; Evensen, 1994) and (c) Monte-Carlo DA
methods. Variational DA minimizes a cost function to obtain the analysis (i.e., the best
estimation of the initial conditions). More specifically, variational DA methods provide a
(quasi) optimal analysis based on an imperfect forecast (*prior state* or *background*), a set of
imperfect observations and their respective error statistics that are prescribed and assumed to
be Gaussian, for simplicity. In addition, variational DA algorithms require a linearized and
adjoint version of the numerical model, which can be very difficult to develop and maintain.
This often involves the use of automatic differentiation tools or complex manual derivation,
both of which are error-prone and time-consuming. On the other hand, the ensemble-based DA
algorithms do not require the use of linearized or adjoint versions of the model, and they do
not use prescribed error statistics. Instead, they compute the error statistics from an ensemble
of forecasts, with the main property that these errors are evolving in time as the system evolves.
The Monte-Carlo DA method allows the assimilation of observations described with non-
Gaussian errors. Particle filters (PF; Van Leewen, 2009; Poterjoy, 2016) are a clear example
of Monte-Carlo DA algorithm. However, PFs are not well-suited for large multidimensional
systems, such as the atmosphere, although a lot of improvements have been achieved recently.
In the present study, we will focus on the most widely used DA schemes typically used in major
operational weather centers, which are the variational and ensemble-based DA schemes,
leaving the Monte-Carlo methods for future work.
Although variational DA schemes have been used in numerical weather prediction for many
years (Courtier et al., 1994; Park and Zupanski, 2003; Rawlins et al., 2007), allowing the
assimilation of a wide range of different observations, they present a well-known limitation.
This limitation is related to the use of a climatological background error covariance matrix to
characterize the error statistics, which is kept constant along the assimilation window, where
the different observations are distributed at different times. This weakness is specifically linked
to the 3DVar method, which typically uses the National Meteorological Center (NMC) method
(Parrish and Derber, 1992) to generate those static background error covariances using forecast
differences over a period of time reasonably close to the event. The error statistics derived from
such DA schemes are static, isotropic and nearly homogenous, misrepresenting the true error
statistics in space and time, which are inherently flow-dependent, resulting in less accurate
analysis. On the other hand, the EnKF DA scheme is designed to provide flow-dependent
background error covariances. Some studies have shown the potential of the EnKF spreading
information from the observations flow-dependently in comparison with the 3DVar (Yang et
al., 2009; Gao et al., 2018). On the other hand, 3DVar techniques require less computational
resources and there is no need to build an ensemble compared to EnKF or even simulate the
model trajectory as in 4DVar. Therefore, the assimilation with 3DVar takes only a few tens of
minutes, making this technique particularly suitable for operational purposes.




To solve convective scale (i.e., grid spacing of a few kilometers) physical processes associated
with extreme weather phenomena, high-resolution numerical simulations are required.
Performing computational expensive high-resolution simulations presents a significant
challenge as it constrains the feasible number of ensemble members that can be used in EnKF
DA schemes, and thus it could hamper significantly the estimation of the background error
covariance matrix. In this context, which DA method is more suitable? The 3DVar using an *ad*
*hoc* background error covariance matrix or the low-rank background error covariance matrix
obtained from the EnKF?
Recently, a few DA studies at convective scale mainly focused just on the mature stage of the
weather event have been carried out (e.g., Wheatley et al., 2015; Jones et al., 2016; Yussouf et
al., 2020). However, investigating the mature stage means that the weather system is already
developed and probably affecting the population. In such situations, the value of improving the
atmospheric condition estimation using DA is very limited in terms of lead time, because there
is no time left for warning the population and to take actions to reduce socio-economic impacts.
In this context, very limited work has been done to assess the impact of DA in pre-convective
systems to significantly improve the lead time, allowing warning systems to act as soon as
possible. Here, we also investigate the role of the 3DVar and EnKF DA methods in improving
pre-convective environment conditions of extreme weather events and how such improved pre-
convective conditions could lead to a forecast improvement with significant time in advance to
warn the population to take actions.
The following study aims at:
(a) Assessing the impact of high-resolution 3DVar in comparison with a high-resolution EnKF
system to predict small-scale extreme weather events initiated over different areas and with
lack of *in-situ* observations.
(b) Investigate the potential of using 3DVar and EnKF to enhance the accuracy of atmospheric
conditions in the pre-convective environment, hours before the mature stage of convective
systems are reached, thereby improving early prediction and warning capabilities for extreme
weather events.
(c) Quantify the impact of assimilating *in-situ* conventional observations in comparison to
assimilating high spatial and temporal resolution data from remote sensing instruments.
(d) Provide a quantitative assessment between the different DA schemes by means of using
several statistical verification methods.

It is important to emphasize that this study is not aimed to draw any statistically significant
conclusion. Instead, we are interested in comparing the performance of EnKF and 3DVar in
two distinct extreme weather events, each with its unique set of conditions and constraints. A
heavy rainfall episode affecting coastal regions of Italy during October 2012 (IOP13; Pichelli
et al., 2017) and a low-predictable Mediterranean Tropical-like cyclone (medicane) affecting
Sicily, known as Qendresa (Pytharoulis et al., 2017; Pytharoulis, 2018; Cioni et al., 2018; Di
Muzio et al., 2019), are used for this study.
This paper is organized as follows. Section 2 briefly describes the meteorological
characteristics of the two events used for comparing the impact of 3DVar and EnKF. In Section
3 the observation dataset that will be assimilated by the different DA methods will be presented.





Section 4 briefly explains the main characteristics of the two DA algorithms that will be used
in this study. Then, the numerical model configuration and the design of the different
experiments for the two different case studies will be described in Section 5 and 6, respectively.
Section 7 describes the verification methods used in this study. Results of the different
numerical experiments for both meteorological situations are summarized in Section 8. Finally,
conclusions are presented in Section 9.

**2. Brief Description of Case Studies**
Two different extreme weather systems, occurring in the Mediterranean region and affecting
populated coastal regions, are considered in this study. The first extreme weather event was
associated with heavy rainfall affecting central and northern Italy during October 2012
(IOP13), while the second extreme weather event was associated with the Qendresa medicane
affecting southern Sicily, Lampedusa, Pantelleria and Malta islands during November 2014.
Both systems were poorly forecasted, and for this reason they are perfect candidates for this
intercomparison study.

**2.1. The IOP13 Heavy Precipitation Episode**
The *IOP13* occurred during the *First Special Observation Period* (SOP1) of the international
project *Hydrological cycle in the Mediterranean Experiment* (HyMeX; Drobinski et al., 2014),
that was mainly designed to better understand heavy rainfall and flash flooding episodes
occurring in the Mediterranean region. The heavy precipitation IOP13 event took place
between 14 and 16 October 2012, and it was characterized by a frontal precipitation system
associated with a deep upper-level trough extending from northern France towards northern
Spain (Fig. 1). It initially affected southern France coastal areas, and afterward it also affected
the northern and central parts of Italy. During 15 October, the Italian rain gauge network
registered 24-hour accumulated precipitation with peaks reaching 60 mm in central Italy, 160
mm in northeastern Italy and 120 mm in Liguria and Tuscany. During the night of 14 October,
a cold front affected the Western Mediterranean region and during 15 October the system
rapidly moved from France to Italy, advecting low-level moisture towards the western coast of
Italy and Corsica, destabilizing the atmosphere and favoring deep moist convective activity.
More details on the synoptic situation and observational data collected during IOP13 can be
found in Ferretti et al., 2014.

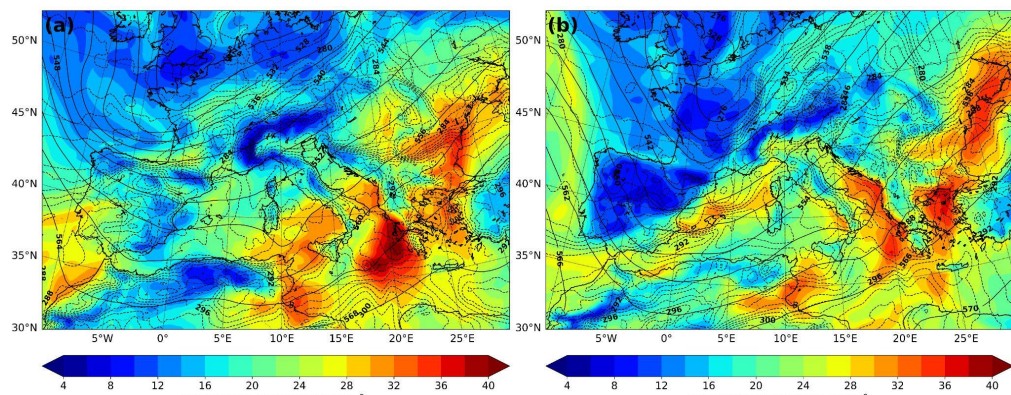

Figure 1. IOP13 ERA5 analyses: 500 hPa geopotential (solid black lines), 925 hPa temperature (dashed grey lines)
and total column of water vapor (color shaded areas) at (a) 12 UTC 14 October and (b) 00 UTC 15 October 2012.

## 2.2. The Qendresa Tropical-Like Cyclone Episode

Among the wide spectrum of maritime extreme weather events, tropical-like Mediterranean
cyclones, a.k.a. medicanes (Emmanuel, 2005), draw particular attention to the community
mainly because they share similar morphological characteristics with tropical cyclones. Given
their tendency to impact densely populated and economically critical areas around the
Mediterranean basin, enhancing the accuracy and reliability of medicanes forecasts has become
an urgent priority. Here, we focus on the 7 October 2014 medicane (Qendresa; Cioni et al.,
2018) that affected the islands of Lampedusa, Pantelleria, Malta and the eastern coast of Sicily.
This event was recognized by the community for its limited predictability (Carrió et al., 2017),
making it a compelling case study for investigating the performance of the 3DVar and EnKF
DA methods. *In-situ* observations located in Malta's airport registered gust wind values
exceeding 42.7 m s$^{-1}$ and a sudden and deep pressure drop greater than 20 hPa in 6 hours.
Satellite imagery during its mature phase showed a well-defined cloud-free eye surrounded by
axisymmetric convective activity, which resembles the morphological properties of classic
tropical cyclones.
A deep upper-level trough associated with a cyclonic flow at mid-levels characterized the
synoptic situation in the Western Mediterranean from 5 to 8 November 2014. The upper-level
trough was associated with an intense PV streamer extending from Northern Europe to
Southern Algeria, and the cyclonic flow at mid-levels was dominated by a strong ridge over
the Atlantic and a deep trough moving along Western Europe. Late on 7 November, the upper-
level trough became negatively tilted, evolving into a deep upper-level cut-off low and the PV
streamer disconnected from the northern nucleus (Fig. 2). A small well-defined spiral-to-
circular cloud shape formed just south of Sicily and evolved east-northeastward, reaching its
maximum intensity over Malta, at midday. Finally, the cyclonic system dissipated as it crossed
the Catania (eastern) coast of Sicily. More details on the synoptic situation and observational
data collected during this event can be found in Carrió et al., 2017.


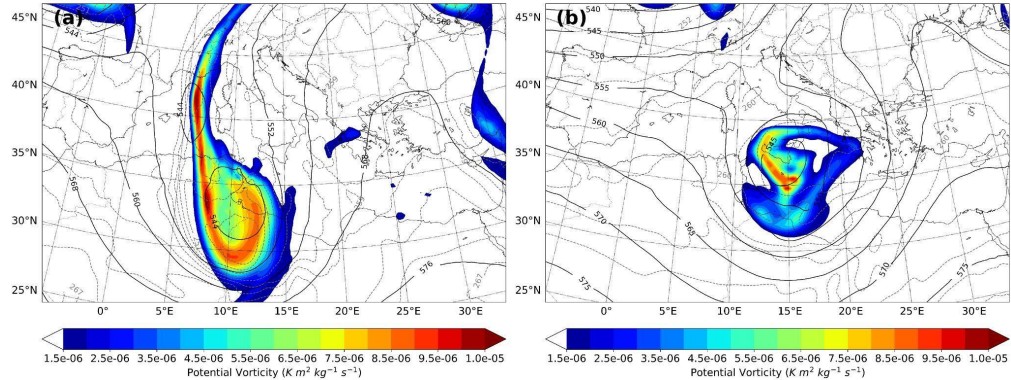

Figure 2. Qendresa ERA5 analyses: 500 hPa geopotential (solid black lines), 500 hPa temperature (dashed grey lines) and 300 hPa Potential Vorticity (color shaded areas) at (a) 00 UTC 7 November and (b) 00 UTC 8 November 2014.

## 3. Observations Description

In this study, different sources of remote-sensing and *in-situ* observations were available for the two case studies. Specifically, the following three types of observations were assimilated: (a) *in-situ* conventional data, (b) high temporal and spatial reflectivity data from two Doppler Weather Radars and (c) 3D wind speed and direction data derived from satellites.

### 3.1. IOP13 Observations

For the IOP13, *in-situ* conventional data and remote sensing observations from two Doppler Weather Radars were available. Moreover, conventional data were obtained from the NOAA's *Meteorological Assimilation Data Ingest System* (MADIS), which has the main advantage of providing high-level quality-controlled data[2] worldwide. In particular, pressure, temperature, humidity and horizontal wind speed and direction from *in-situ* instruments such as METARs, maritime buoys, rawinsondes and aircrafts (Fig. 3a). In addition to these conventional observations, reflectivity data from two Météo-France polarimetric S-band Doppler Weather Radars, were also available on the Gulf of Genoa. One located in Corsica Island (9.496ºE, 42.129ºN) at 63 m ASL, known as Aleria, and the other located in southern France (4.502ºE, 43.806ºN) at 76 m ASL, known as Nimes (Fig. 3a). These two radars, strategically positioned, ensure a good spatial coverage over the Ligurian Sea, the area where initiation and intensification of deep convection occurred, and provide key information about the 3D structure of the convective systems at high spatial and temporal resolution. The two radars perform 5 and 9 elevation scans every 5 minutes, respectively, and their data are available at the HyMeX's official website (see https://www.hymex.org). Specifically, Aleria radar provides data at 5 elevation angles: 0.57º, 0.96º, 1.36º, 3.16º and 4.57º with a mean frequency of 2.8 GHz. In comparison, Nimes radar provides data at 9 elevation angles: 0.58º, 1.17º, 1.78º, 2.38º, 3.49º, 4.99º, 6.5º, 7.99º and 89.97º, also at the same frequency. It is worth mentioning that Aleria and Nimes radar reflectivity data are provided by the Météo-France operational radar network and undergo rigorous data quality control. This ensures that common radar error sources, such as signal attenuation, ground clutter or beam blocking, are meticulously identified

---

[2] See https://madis.ncep.noaa.gov/madis_qc.shtml for further details on the Quality Control techniques used.




and corrected. Radial velocity from Aleria and Nimes Doppler radars was also available, but
because of the low reliability of the data (not quality controlled properly) it was not used in this
study.
Hence, the following observations were assimilated for this event:
• Conventional *in-situ* data were hourly assimilated over the entire numerical domain
considered (Fig. 3a).
• Reflectivity data from two weather radar from Météo-France were assimilated every 15
minutes (Fig. 3a).
The high spatial resolution of the reflectivity data poses significant challenges for their direct
assimilation, potentially leading to detrimental analysis related with signal aliasing and the
violation of the uncorrelated observational error assumptions followed in the derivation of the
3DVar and EnKF analysis equations. To mitigate the adverse effects associated with these
issues, the *Cressman Objective Analysis* technique (Cressman, 1959) was used to interpolate
raw radar observations to a regularly spaced 6 km horizontal grid, as suggested by previous
work (i.e., Wheatley et al., 2015; Yussouf et al., 2015). It is important to note that reflectivity
observations are typically obtained in polar coordinates, a prerequisite step before applying the
Cressman interpolation involves converting them to a Cartesian coordinate system. We have
performed several sensitivity tests using different grid space resolution (e.g., 3, 6, 9 km) and
we found that using 6 km grid space produces the best analysis. To reduce spurious convective
signals and remove excessive humidity the *null-echo* option, which allows assimilation of no
precipitation echoes, has been adopted in 3DVAR experiment.

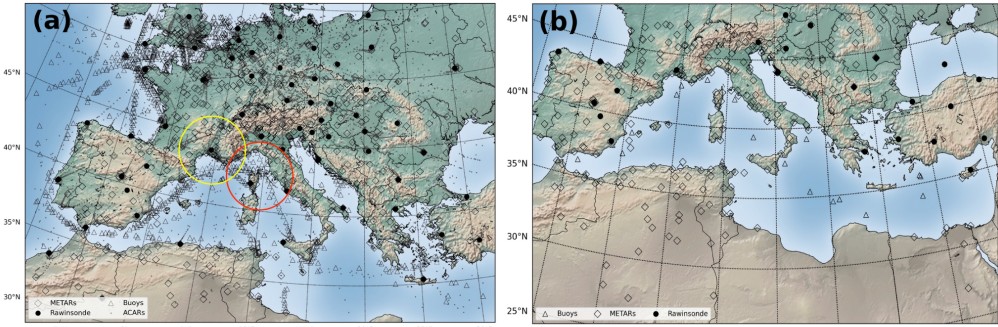

Figure 3. (a) IOP13 Episode: Spatial distribution of *in-situ* observations (gray and black markers) assimilated on
the parent numerical domain during 24 h assimilation window from 00 UTC 14 October to 00 UTC 15 October
2012. Doppler Weather Radars located at Nimes and Aleria and their coverage range, depicted in yellow and red
circles, respectively. (b) Qendresa Episode: Spatial distribution of *in-situ* observations hourly assimilated during
12 h assimilation window from 12 UTC 6 November to 00 UTC 7 November 2014.

**3.2. Qendresa Observations**
For the Qendresa episode, two different observational sources were available: (a) conventional
*in-situ* observations and (b) satellite-derived observations. Conventional *in-situ* observations
were obtained from MADIS database. However, only observations from buoys, METAR and
rawinsonde were used for this case. It is essential to highlight that observation gaps persist





across large areas of the region, particularly over the sea (Fig. 3b), where Qendresa initiated
and evolved. As for the IOP13, we were interested in Doppler Weather Radars data to enhance
the intensity and trajectory forecasts of Qendresa. Unfortunately, Doppler Weather Radars
were not available in the neighborhood of the region where Qendresa initiated and evolved,
but another source of observations, the so-called *Rapid-Scan Atmospheric Motion Vectors*
(RSAMVs; Velden et al., 2017), which provides 3D wind information throughout the entire
atmosphere (both speed and direction) at high spatial and temporal resolution (i.e., every 20-
min), were available for this event over the sea. This satellite product is obtained using the
*Spinning Enhanced Visible and Infrared Imager* (SEVIRI) instrument onboard the *Meteosat*
*Second Generation* (MSG) satellite, which has a scanning frequency as low as 5 minutes. The
final product is indeed obtained averaging 4 consecutive images.
Hence, the following observations were assimilated for this event:
● Conventional *in-situ* data from buoys, METAR and rawinsonde for the entire
Mediterranean region were hourly assimilated.
● Wind speed and direction from the *Rapid-Scan Atmospheric Motion Vectors* for the
entire atmosphere at high spatial and temporal resolution were assimilated every 20
minutes.
Recent studies have shown that upper-level dynamics played a key role in the genesis and the
development of Qendresa (Carrió et al., 2017; Carrió, 2022), so the assimilation of RSAMVs
is expected to significantly improve its predictability. Here, the infrared channel from
RSAMVs (10.8 $\mu m$), which contains information throughout the entire atmosphere, was
selected to be assimilated (Fig. 4). However, before assimilating RSAMVs, a quality control
check to reject non-physical and outlier observations that could deteriorate the quality of the
analysis and the successive forecast was applied. In addition, to minimize the effect of having
spatial correlated observation errors associated to high density observations, the "*superobbing*"
technique consisting in reducing the data density through spatially averaging the observations
within a predefined prism is applied (i.e., Pu et al., (2008); Romine et al., (2013); Honda et al.,
(2018)). Based on the most accurate analysis obtained by multiple sensitivity experiments (not
shown) for Qendresa, the RSAMVs data are thinned using a prism with horizontal dimensions
of 128x128 km$^2$ and 25 hPa in the vertical dimension.

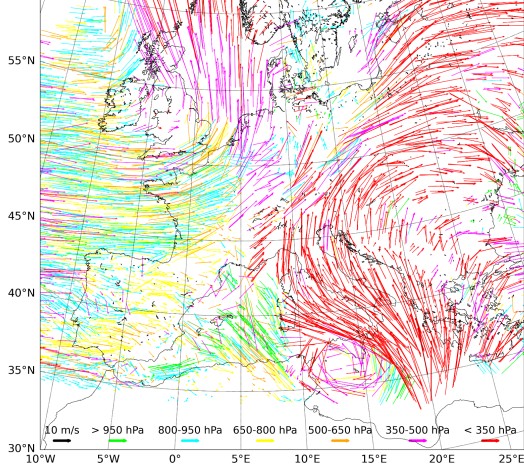




Figure 4. Raw EUMETSAT's RSAMV observations depicted at different vertical levels by infrared channel 10.8 *μm* at 12 UTC on 7 November 2014 over the Mediterranean region. Wind information is only valid at the center of the wind vectors.

Observations from aircraft (i.e., ACARS) were not assimilated in this case because preliminary assimilation tests indicated a worsening of the results and led to a poorer estimation of the atmospheric state. Buoys, METAR and rawinsonde observations covering the entire Mediterranean region were hourly assimilated.

Finally, observational errors used for the assimilation of the observations associated with both IOP13 and Qendresa are motivated by Table 3 in Romine et al., (2013) with the following minor changes: METAR altimeter (1.5 hPa), marine altimeter (1.20 hPa), METAR and marine temperature (1.75 K) and RSAMV wind observations (1.4 m s$^{-1}$). These minor changes are found to provide better data assimilation analysis for the IOP13 and Qendresa extreme weather events in the Mediterranean region. The remaining of the observation errors are the same as the ones in Romine et al., (2013).

## 4. Data Assimilation Schemes

In the present study, two widely used data assimilation algorithms are used for improving the forecast of extreme weather events initiated and developed over poorly observed maritime regions and affecting densely populated coastal areas. We refer to the *Ensemble Adjustment Kalman Filter* and the variational *3DVar* data assimilation schemes, which are described below.

### a) The Ensemble Adjustment Kalman Filter (EnKF)

The *Ensemble Adjustment Kalman Filter* (EAKF; Anderson 2001), which is implemented in the *Data Assimilation Testbed Research* (DART[3]), is used in this study as the former ensemble-based data assimilation technique. The EAKF provides an optimal estimation, in the least square error sense, of the true probability distribution of the state of the atmosphere by merging two main sources of information: (a) the available observations and (b) an ensemble of forecasts (a.k.a. *background*) valid at the analysis time. In particular, the EAKF assimilates the observations serially. This means that the analysis ensemble obtained by the EAKF after the assimilation of the first observation at a given time is then used as the *background* for the next observation at the same analysis time. This is done recursively until all the observations valid at the same analysis time are finally assimilated.

---

[3] http://www.image.ucar.edu/DAReS/DART/



In particular, for each observation $j$ from a set of $p$ observations valid at the same analysis time,
the EAKF can be summarized with the 4 main steps described below:

**Step 1)** Obtain the observed value $y_j^o$, and the associated observation error variance, $R_{jj}$
**Step 2)** Update the ensemble mean $\langle y_j^f \rangle$ and ensemble members $y_j^f$ of the observed variable
using:

$$\langle y_j^a \rangle = \langle y_j^f \rangle + \frac{H_j Z \left( H_j Z \right)^T}{H_j Z \left( H_j Z \right)^T + R_{jj}} \left( y_j^o - \langle y_j^f \rangle \right) = \langle y_j^f \rangle + \frac{H_j \mathbf{P}^f H_j^T}{H_j \mathbf{P}^f H_j^T + R_{jj}} \left( y_j^o - \langle y_j^f \rangle \right) \qquad \text{(Eq. 1)}$$

$$y_{ji}^a = y_{ji}^a + \sqrt{\left[ 1 - P^f \left( P^f + R_{jj} \right)^{-1} \right]} \left( y_{ji}^f - \langle y_j^f \rangle \right); \text{ for } i=1,...,K \text{ and } P^f = H_j P^f H_j^T \qquad \text{(Eq. 2)}$$

where $y_{ji}^f = H_j \left( \mathbf{x}_i^f \right); \ \langle y_j^f \rangle = \frac{1}{K} \sum_{i=1}^{K} H_j \left( \mathbf{x}_i^f \right); \ H_j Z = \frac{1}{\sqrt{K-1}} \left[ y_{j1}^f - \langle y_j^f \rangle, ..., y_{jK}^f - \langle y_j^f \rangle \right]$

**Step 3)** Find corresponding analysis ensemble for the observations and model variables using
a linear regression step:

$$y_{ki}^a = y_{ki}^f + \frac{H_k Z \left( H_j Z \right)^T}{H_k Z \left( H_j Z \right)^T + R_{jj}} \left( y_{ji}^a - y_{ji}^f \right), \text{ for } k = 1,...,p \text{ and } i=1,...,K \qquad \text{(Eq. 3)}$$

$$x_{\mu i}^a = x_{\mu i}^f + \frac{Z \left( \mu, : \right) \left( H_j Z \right)^T}{H_k Z \left( H_j Z \right)^T + R_{jj}} \left( y_{ji}^a - y_{ji}^f \right), \text{ for } \mu = 1,...,n \text{ and } i=1,...,K \qquad \text{(Eq. 4)}$$


where $n$ is the number of model variables.
**Step 4)** Let the analysis ensemble become the background ensemble for the next observation:

$$y_{ki}^f = y_{ki}^a, \quad \text{for } k = 1,...,p \text{ and } i=1,...,K \qquad \text{(Eq. 5)}$$
$$x_{\mu i}^f = x_{\mu i}^a, \quad \text{for } \mu = 1,...,n \text{ and } i=1,...,K \qquad \text{(Eq. 6)}$$

In the above equations, $K$ is the number of ensemble members, $p$ the number of observations,
$n$ is the number of model variables, $H$ is the observation operator (non-linear) and $Z$ the
ensemble perturbations about the mean. The superscripts "$a$" and "$f$" stand for the analysis and
forecast, respectively.
Ensemble covariances used in high-resolution simulations, such as the present study, where
only a limited number of ensemble members is feasible, suffers from sampling error, resulting
in the generation of spurious correlations that hamper the analysis (Hacker et al., 2007). The
detrimental effects of these spurious correlations are mitigated by employing covariance
localization functions that go zero as the distance between the assimilated observation and
the grid model point where the analysis occurs, increases (Houtekamer and Mitchell, 1998). In
our case, a fifth-order piece-wise rational Gaussian localization function is used (Gaspari and
Cohn, 1999). For this study, after several sensitivity simulations it was found that using a half-
radius[4] of 230 km in the horizontal and a half-radius of 4 km in the vertical for the horizontal
and vertical localizations, respectively, results in the best performance of the DA scheme.
The assimilation of each observation results in a reduction of the ensemble spread, attributed
to using a reduced-moderate ensemble size (Anderson and Anderson, 1999). To address this
issue and help to maintain the spread, an *adaptive inflation technique* (Anderson and Collins,
2007; Anderson et al., 2009) is applied to the prior ensemble before assimilating the
observations. The adaptive inflation technique increases the spread of the ensemble without
changing the mean. The inflation value has a probability density distribution described by a
mean and a standard deviation. In this study, it was determined that initializing the mean value
of inflation at 1.0 and using a standard deviation of 0.6, yields the best performance of the DA
scheme.

**b) Three-dimensional Variational Data Assimilation (3DVar)**
The 3DVar technique, implemented in WRFDA (Barker et al., 2004), is adopted for the
numerical simulations. The 3DVar aims to seek the best estimate of the initial conditions
through the iterative minimization of a cost function:

$$J(\mathbf{x}) = \frac{1}{2}\left\{ (\mathbf{x}-\mathbf{x}_b)^T \mathbf{B}^{-1}(\mathbf{x}-\mathbf{x}_b) + \left[\mathbf{y}_o - \mathbf{H}(\mathbf{x})\right]^T \mathbf{R}^{-1}\left[\mathbf{y}_o - \mathbf{H}(\mathbf{x})\right] \right\} \qquad \text{(Eq.7)}$$


where $\mathbf{B}$ and $\mathbf{R}$ are the background and observation error matrices, respectively; $\mathbf{x}$ is the state
vector; $\mathbf{y}_o$ is the observations, $\mathbf{x}_b$ is the first guess and $H$ is the forward (non-linear) operator
that converts data from model space to observation space.
The solution of the above cost function J consists in finding a state $\mathbf{x}_a$ (analysis), that minimizes
the distance between the observations and the background field. However, in a model with $10^6$
degrees of freedom, the direct solution is computationally expensive. To reduce the complexity
and calculate $\mathbf{B}^{-1}$ more efficiently, a pre-conditioning is applied by transforming the control
variables, respectively, pseudo relative humidity, temperature, u, v, and surface pressure, as $\mathbf{x}$
$- \mathbf{x}_b = \mathbf{U}v$, where v is the control variable and $\mathbf{U}$ the transformation operator.
Regarding the assimilation of radar reflectivity, the observation operator from Sun and Crook
(1997) is adopted:

$$Z = a + b\log_{10}(\rho q_r) \qquad \text{(Eq. 8)}$$


where $Z$ is the reflectivity, $q_r$ is the rainwater mixing ratio, $\rho$ the air density whereas the
coefficients $a$ and $b$ are equal to 43.1 and 17.5, respectively.
The background error covariance matrix $\mathbf{B}$ matrix plays a key role in the assimilation process
by weighing and smoothing the information from observations and by ensuring a proper
balance between the analysis fields. The *National Meteorological Center* method (NMC;
Parrish and Derber, 1992) was used to model the B matrix. This method evaluates the

---

[4] The half-radius or cutoff term is defined here as 0.5 times the distance to where the impact of the observation assimilated go to zero. Multiplying the half-radius by 2 results in the maximum distance at which an observation can modify the model state.




differences, over a period of two weeks, between two short-term forecasts valid at the same
time but with different lead time, 12h and 24h, respectively, to generate the forecast error
covariance matrix **B**. Recently, several works (Wang et al., 2013; Li et al., 2016; Shen et al.,
2022; Ferrer Hernandez et al., 2022) show the benefit of using a slightly different approach for
the **B** matrix (CV7) in assimilating radar reflectivity, besides in precipitation forecast accuracy.
The CV7 differs from the others by using empirical orthogonal functions (EOFs) to represent
the vertical covariance instead of a vertical recursive filter. Moreover, the control variables are
in eigenvector space, and they are the following: u, v, temperature, pseudo relative humidity
($RH_s$), and surface pressure ($P_s$). Therefore, CV7 option has been used to generate the **B** matrix
for both case studies. In this study, the weak penalty constraint (WPEC) option (Li et al., 2015)
implemented in WRFDA has been activated to improve the balance between the wind and
thermodynamic state variables, enforcing the quasi-gradient balance on the analysis field.

**5. Model set-up**
The mesoscale Advanced Research Weather Research and Forecasting Model (WRF;
Skamarock et al., 2008) version 3.7 is used in this study. WRF solves a fully compressible and
non-hydrostatic set of equations, using a $\eta$ terrain-following hydrostatic-pressure vertical
coordinate. The Arakawa C-grid staggering scheme and a third-order Runge-Kutta time-
integration, to improve the precision of the numerical solutions, are used. Because IOP13 and
Qendresa episodes took place in different locations and with different conditions, two different
model configurations were used. For the IOP13 episode, a one-way nested model configuration
with the parent domain centered over the Western Mediterranean Sea, covering Central Europe
and North Africa, with a horizontal grid-resolution of 15 km (168x247) and a nested domain
centered over Gulf of Genoa with a horizontal grid-resolution of 3 km (250x250) were used
(Fig. 5a). Both domains were characterized to have 51 vertical model levels, from surface to
50 hPa, with higher density of levels in the lower part of the atmosphere than in the upper. For
Qendresa, a two one-way nested model configuration is also used, but now the parent domain
is centered over the Central Mediterranean Sea, covering most of the European region and the
northern part of Africa (Fig. 5b), using a horizontal grid resolution of 15 km (245x245). The
nested domain is centered over Sicily (Southern Italy) using a grid resolution of 3 km
(253x253). Both numerical domains use a 51 terrain-following $\eta$ levels up to 50 hPa, as in the
IOP13 case.
For the EnKF DA experiments, initial and boundary conditions used to perform the simulations
associated with IOP13 were obtained from the *European Center of Medium Range Weather*
*Forecasts Global Ensemble Prediction System* (EPS-ECMWF), which stored meteorological
fields using a horizontal and vertical spectral triangular truncation of T639L62 (i.e., ~32 km
grid resolution in the horizontal). In particular, the EPS-ECMWF provides 51 different initial
and boundary conditions from 50 perturbed ensemble members plus a control simulation.
However, due to unfeasible computational resources required to run our numerical simulations
at high grid resolution, here we will use an ensemble consisting of 36 members. This
configuration is analogous to the one used at the internationally prestigious *National Oceanic*
*and Atmospheric Administration - National Severe Storms Laboratory* (NOAA-NSSL) in
Norman (Oklahoma, USA) to improve predictability of tornadoes. To obtain the desired 36-
member ensemble, a *Principal Components Analysis* and *K-mean* clustering technique were
used together to select the 36 ensemble members from the EPS-ECMWF showing more
dispersion over the entire numerical domain (see Garcies and Homar, 2009 and Carrió et al.,
2016 for more details using these techniques). To perform Qendresa DA simulations, the initial




and boundary conditions were obtained following the same methodology explained above for
the IOP13 case, i.e., using an ensemble of 36 members obtained from the EPS-ECMWF. On
the other hand, the initial and boundary conditions for 3DVar simulations are provided by the
*Integrated Forecast System* (IFS) global model from the ECMWF, with a spatial resolution of
0.1° x 0.1° and updated every 3 hours.

Figure 5. Mesoscale and storm-scale numerical domains used in this study for the (a) IOP13 and (b) Qendresa
episodes, respectively.




To estimate the uncertainties of WRF, which is a necessary information for the EnKF, a
multiphysics ensemble is built for both the IOP13 and Qendresa event (e.g., Stensrud et al.,
(2000); Wheatley et al., (2012)), where each ensemble member gets a different set of
parameterizations (see Table 1). In particular, the diversity in our ensemble consists of (a) two
short- and long-wave radiation schemes [Dudhia (Dudhia, 1989) and RRTMG (Iacono et al.,
2008)], (b) three cumulus parameterizations schemes [Kain-Fritsch (KF; Kain and Fritsch,
1993; Kain, 2004), Tiedtke (Tiedtke, 1989) and Grell-Freitas (GF; Grell and Freitas, 2013)]
and (c) three planetary boundary layer schemes [Yonsei University (YSU; Hong et al., 2006),
Mellor-Yamada-Janjic (MYJ; Janjic, 1990, 2001), and Mellor-Yamada-Nakanishi-Niino level
2.5 (MYNN2; Nakanishi and Niño, 2006, 2009)]. Two widely used physics parameterizations
are adopted for the microphysical processes and land surface interactions, the New Thompson
(Thompson et al., 2008) and Noah (Tewari et al., 2004) schemes, respectively. Note that the
above-mentioned physical parameterizations are used for both the large-scale ensemble in the
parent domain and the storm-scale ensemble in the nested domain, except for the cumulus
parameterization that is only applied in the parent domain ensemble. On the other hand, for the
WRF deterministic simulation using 3DVar, the microphysical processes are parametrized by
using the New Thompson scheme, while a YSU scheme is adopted for PBL. Long- and short-
wave radiation are considered through a RRTMG and Dudhia scheme, respectively; while
Kain-Fritsch scheme is used for the convection, except for the inner domain where it is
explicitly resolved.

**Table 1**: Multiphysics parameterizations used to generate the 36-member ensemble for the EnKF experiments in
IOP13 and Qendresa episodes. PBL, SW and LW stand for planetary boundary layer, short-wave and long-wave,
respectively.

| **Multiphysic Configuration** | | | | | | | | | | |
|---|---|---|---|---|---|---|---|---|---|---|
| **Ens. Memb.** | **MP** | **CU** | **PBL** | **Land Sfc** | **SW/LW Rad.** | **Ens. Memb.** | **MP** | **CU** | **PBL** | **Land Sfc** | **SW/LW Rad.** |
| 1 | New Thompson | KF | YSU | Noah | Dudhia | 19 | New Thompson | KF | YSU | Noah | Dudhia |
| 2 | New Thompson | KF | YSU | Noah | RRTMG | 20 | New Thompson | KF | YSU | Noah | RRTMG |
| 3 | New Thompson | KF | MYJ | Noah | Dudhia | 21 | New Thompson | KF | MYJ | Noah | Dudhia |
| 4 | New Thompson | KF | MYJ | Noah | RRTMG | 22 | New Thompson | KF | MYJ | Noah | RRTMG |
| 5 | New Thompson | KF | MYNN2 | Noah | Dudhia | 23 | New Thompson | KF | MYNN2 | Noah | Dudhia |
| 6 | New Thompson | KF | MYNN2 | Noah | RRTMG | 24 | New Thompson | KF | MYNN2 | Noah | RRTMG |
| 7 | New Thompson | GF | YSU | Noah | Dudhia | 25 | New Thompson | GF | YSU | Noah | Dudhia |
| 8 | New Thompson | GF | YSU | Noah | RRTMG | 26 | New Thompson | GF | YSU | Noah | RRTMG |
| 9 | New Thompson | GF | MYJ | Noah | Dudhia | 27 | New Thompson | GF | MYJ | Noah | Dudhia |




| 10 | New Thompson | GF | MYJ | Noah | RRTMG | 28 | New Thompson | GF | MYJ | Noah | RRTMG |
| 11 | New Thompson | GF | MYNN2 | Noah | Dudhia | 29 | New Thompson | GF | MYNN2 | Noah | Dudhia |
| 12 | New Thompson | GF | MYNN2 | Noah | RRTMG | 30 | New Thompson | GF | MYNN2 | Noah | RRTMG |
| 13 | New Thompson | Tiedke | YSU | Noah | Dudhia | 31 | New Thompson | Tiedke | YSU | Noah | Dudhia |
| 14 | New Thompson | Tiedke | YSU | Noah | RRTMG | 32 | New Thompson | Tiedke | YSU | Noah | RRTMG |
| 15 | New Thompson | Tiedke | MYJ | Noah | Dudhia | 33 | New Thompson | Tiedke | MYJ | Noah | Dudhia |
| 16 | New Thompson | Tiedke | MYJ | Noah | RRTMG | 34 | New Thompson | Tiedke | MYJ | Noah | RRTMG |
| 17 | New Thompson | Tiedke | MYNN2 | Noah | Dudhia | 35 | New Thompson | Tiedke | MYNN2 | Noah | Dudhia |
| 18 | New Thompson | Tiedke | MYNN2 | Noah | RRTMG | 36 | New Thompson | Tiedke | MYNN2 | Noah | RRTMG |


## 6. Design of IOP13 and Qendresa Experiments

To quantitatively assess the benefits of assimilating different types of observations using the 3DVar and the EnKF DA schemes, a few numerical experiments are performed. A reference experiment without any data assimilation is carried out. Then, several numerical experiments using different types of observations for the assimilation are performed. Only conventional *in-situ* observations are assimilated using the 3DVar and the EnKF, for the first set of experiments. All available observations (i.e., conventional, radar based and satellite derived data) are assimilated using both 3DVar and EnKF, for the second type of experiments. The comparison between these numerical experiments will provide information on which DA scheme and observation is performing better for these weather events. The DA experiments mainly consist of two phases: the first one is related to the data assimilation procedure, where different types of observations are assimilated by the variational 3DVar and the ensemble-based EnKF DA schemes; the second phase is associated with the free model run initialized using the initial conditions obtained during the first phase. The total forecast time is 24 h and 36 h for IOP13 and Qendresa, respectively. For IOP13, a further simulation lasting 6-hour from 18 UTC 13 October to 00 UTC 14 October 2012 (Carrió et al., 2019) is performed (Fig. 6) to reduce spin-up problems related to the direct downscaling from global ECMWF analysis (32 km grid resolution) to the WRF parent domain used in our simulations (16 km grid resolution). This procedure improved the DA for IOP13, but it had a small impact for Qendresa.

Therefore, the following model simulations were performed:

- No Data Assimilation (NODA)

- Only conventional *in-situ* observations are assimilated using the 3DVar and the EnKF (SYN)

- All available observations (i.e., conventional, radar based and satellite derived data) are assimilated using both 3DVar and EnKF (CNTRL)




The comparison between SYN and CNTRL will allow for assessing the role of radar and/or
satellite data, especially for the events originated in the area where observations are not
available. Moreover, the assimilation of the radar and/or satellite will produce important
information on the triggering phase of both events developing on the sea.

## 6.1. CNTRL Experiments

For IOP13, the CNTRL experiment is designed to assimilate both *in-situ* conventional and
reflectivity observations from Aleria and Nimes Doppler weather radars. The assimilation of
the reflectivity is expected to improve the forecast of this event by significantly improving the
initial conditions over the sea, where convective activity initiated and evolved into deep
convection affecting coastal populated areas of Italy. As briefly described in the previous
section, this experiment consists of three stages: 1) the spin-up of the storm-scale domain is
accounted for by running the WRF model during 6 hours from 18 UTC 13 October to 00 UTC
14 October 2021. Note that for the 3DVar experiment, the spin-up is accounted by just
initializing WRF with the deterministic analysis from the IFS ECMWF. However, for the EnKF
counterpart, the spin-up is accounted by initializing the 36-member ensemble at 18 UTC 13
October; 2) *in-situ* conventional observations were hourly assimilated during 24 hours from 00
UTC 14 October to 00 UTC 15 October, meanwhile reflectivity observations were assimilated
using a Rapid-Update Assimilation Cycle every 15 minutes during a period of 6 hours, from
18 UTC 14 October to 00 UTC 15 October (Fig. 6b); and 3) a 24-h ensemble (deterministic)
forecast until 00 UTC 16 October, using the recently obtained initial conditions, is performed
by the EnKF (3DVar).
For the Qendresa episode, CNTRL experiment is designed to assimilate both *in-situ*
conventional and RSAMV observations. The assimilation of RSAMV observations is expected
to improve the representation of the atmospheric circulation at upper-levels, whereas the
assimilation of surface conventional observations is expected to enhance the one at low-levels.
The Qendresa CNTRL experiment consists of two main phases: 1) *in-situ* conventional and
satellite derived RSAMV observations are hourly and 20-min assimilated, respectively, during
a 12-h period from 12 UTC 6 November to 00 UTC 7 November 2014 to end up with the last
analysis at the end of the assimilation window (i.e., 00 UTC 7 November); 2) a free 36-h
ensemble (deterministic) forecast is performed by the EnKF (3DVar) from 00 UTC 7
November to 12 UTC 8 November 2014 (Fig. 6e).

## 6.2. SYN Experiments

For IOP13, the SYN experiment assesses the impact of *in-situ* conventional observations,
which are crucial to characterize mesoscale atmospheric circulation. Analogous to the CNTRL,
SYN follows the same three phases, but in the second phase only the hourly *in-situ*
conventional observations from 00 UTC 14 October to 00 UTC 15 October 2012 are
assimilated. The analysis obtained from the assimilation stage is used as initial conditions for
running the free forecast for 24h, in the third phase (Fig. 6a).
Similarly, also for Qendresa, in the SYN experiment only *in-situ* conventional observations are
hourly assimilated for 12 hours, from 12 UTC 6 November to 00 UTC 7 November 2014 (Fig.
6d).



### 6.3. NODA Experiments

For the IOP13, NODA experiment is a direct downscaling from EPS-ECMWF boundary and
initial conditions valid at 00 UTC 15 October to 00 UTC 16 October 2012. To the aim of
simulating an operational framework, the NODA experiment starts at 00 UTC 15 October,
instead of starting at 18 UTC 14 October (Fig. 6c). With this choice of the starting time, one
could answer the question of which forecast system we should use to predict a 24-48 h forecast.
Should we simply perform a simple downscaling using the last analysis obtained from a global
model, or should we start our simulation with a previous analysis but now using DA at high
temporal and spatial resolution to enhance the estimation of the initial conditions? The
comparison among NODA, CNTRL and SYN will provide us with valuable information on the
impact of assimilating different sources of observations.
For Qendresa, NODA experiment is simply a direct downscaling of 36 hours from EPS-
ECMWF at 00 UTC 7 November to 12 UTC 8 November 2014 (Fig. 6f). Here again, it is
important to note that the choice of starting NODA at 00 UTC 7 November instead of starting
at 12 UTC 6 November was made intentionally to extract general conclusions applicable to an
operational framework.

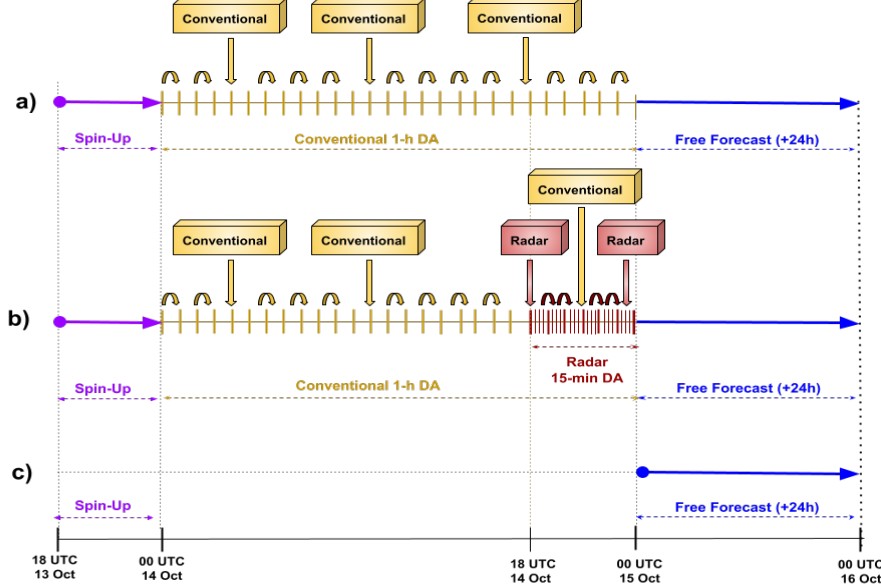

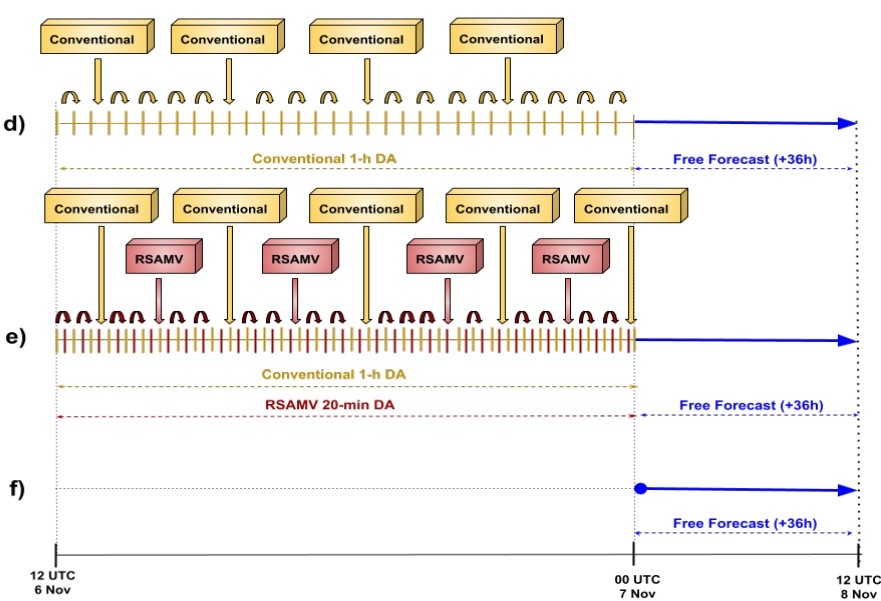

Figure 6. Schematic representation of the main numerical experiments performed in this study for the IOP13 and Qendresa episodes, respectively. SYN, CNTRL and NODA experiments for the IOP13 are shown in (a), (b) and (c) panels, respectively, meanwhile the ones corresponding to Qendresa are shown in (d), (e) and (f), respectively.

## 7. Verification Methods

To quantitatively evaluate the performance of the EnKF and the 3DVar and their impact on the short-term forecasting of these two extreme weather events, various verification scores are used. Given the different nature of the weather phenomena associated with these episodes, the selection of verification scores is tailored specifically to each event. For the IOP13 heavy precipitation event (Fig. 7a), the model verification was performed using the observed accumulated precipitation field over different time windows (e.g., 3 hours, 6 hours or 9 hours). More specifically, the accumulated precipitation was computed using observations from the *Italian Department of Civil Protection*. However, the spatial distribution of rain gauges is not homogenous and there are regions where a lack of rain gauges is present. To address these issues, three sub-regions are chosen where the heavy precipitation event was well recorded by the weather stations (see R1, R2 and R3 in Fig. 7b). Conversely, for the Qendresa tropical-like cyclone, a limited number of *in-situ* observations were present since it initiated and moved over the sea during its lifecycle, and radar-data were not available. Consequently, IR satellite imagery was the primary source of data to approximately estimate Qendresa's trajectory (Fig. 7c). Regarding the intensity of Qendresa, since the cyclone's center passed over Malta island, reaching its minimum mean sea level pressure (MSLP) of 985 hPa, METAR data from Malta's airport was also used to verify the cyclone's intensity (Fig. 7d).


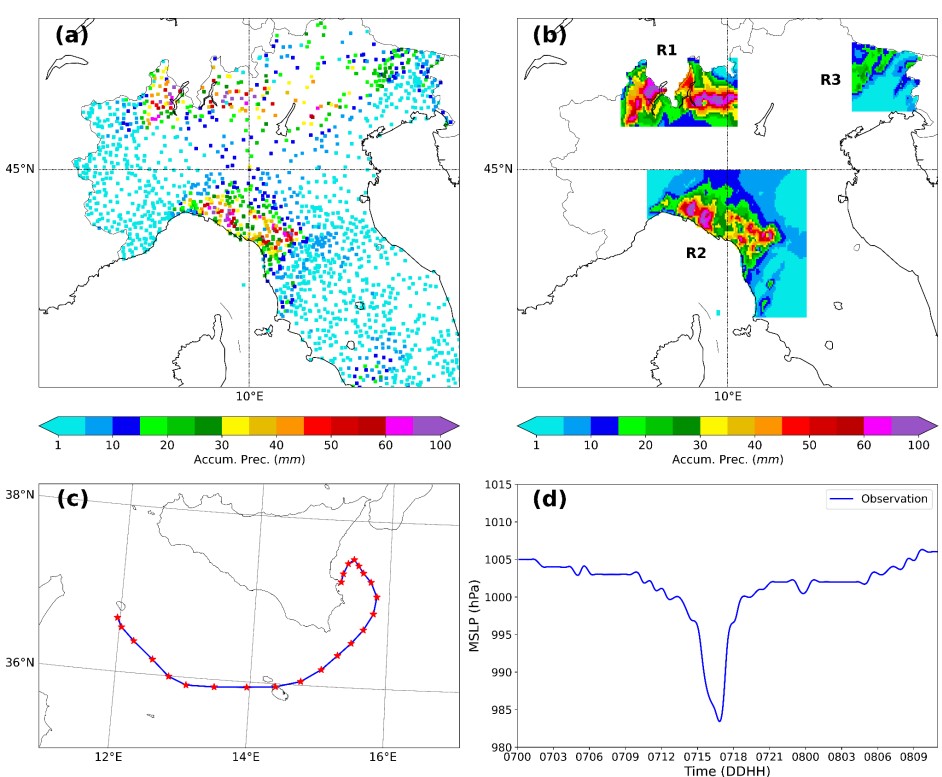

Figure 7. (a) Example of the 12-h accumulated precipitation estimated values and their spatial distribution from the *Italian Department of Civil Protection* rain gauges. (b) Linear interpolation of 12-h accumulated precipitation values into the three target areas where verification has been performed. (c) Observed track of Qendresa medicane viewed from infrared satellite imagery. (d) Surface pressure (hPa) data obtained from the METAR station at Malta's airport.

To quantitatively assess the short-term (i.e., first 6-9 hours) precipitation forecast for the IOP13 initialized using the analysis from the 3DVar and EnKF DA techniques, the *Filtering Method*, the *Relative Operating Characteristics* (ROC; Mason, 1982; Stanski et al., 1989; Swets, 1973) and the *Taylor Diagrams* (Taylor, 2001) were used. We avoid using the conventional point-by-point approach, which has been shown to have serious limitations in the evaluation of high-grid spatial and temporal precipitation field resolutions (Roberts, 2003). More specifically, as *Filtering Method* we use the *Fraction Skill Score* (FSS; Roberts and Lean, 2008), which is commonly used to quantitatively assess precipitation. A preliminary interpolation of the forecast and the observations onto a common regular mesh of 3 km is performed to compute FSS. Then the comparison is carried out within a region of 3x3 grid cells around each grid cell. The FSS can be used to determine the scale over which a forecast system has sufficient skill (Mittermaier, 2010). The FSS ranges from 0 to 1, being 1 a perfect match between model and observations. In addition to the ROC curves, the *Area Under the ROC Curve* (AUC; Stanski et al., 1989; Schwartz et al., 2010), which is also widely used to quantitatively assess the quality of weather forecasts, will be also used in this study. For a perfect forecast, AUC is equal to 1.

For Qendresa, the *Whisker diagrams* (Tukey, 1977) and the *Probability Distribution of the Cyclone Center Occurrence* **(PCCO)**, which was based on the *Kernel Density Estimation*


(KDE; Bowman and Azzalini, 1997; Scott, 2015; Silverman, 2018), were used to validate the
simulations. More specifically, the KDE is used to compute the probability of having the center
of the cyclone over the entire numerical domain. The main idea behind KDE is to place a
"kernel" (i.e., a probability distribution function) at each data point, and then sum up the kernels
to estimate the overall probability density function. The kernel is typically chosen to be a
smooth function, such as a Gaussian, that decays to zero as the distance from the data point
increases. The width of the kernel is controlled by a parameter called the bandwidth, which it
turns out to be one of the limitations of the KDE technique. In this case, we found that the
optimal bandwidth is 20 km, which is within the meso $\beta$ scale, i.e. a typical length scale for
convective cells. Here, a 2-dimensional KDE will be applied over each cyclone center (*lat*, *lon*
coordinates) identified for the different simulations (i.e., EnKF vs 3DVar). In this way, we will
infer the most probable track of Qendresa for the different simulations, thereby identifying
which is the best DA technique and which provides better estimations of Qendresa medicane's
track.

### 8. Results

To quantitatively estimate the impact on the short-range forecast from assimilating the different
types of observations considered in this study, using the 3DVar and the EnKF, the
abovementioned verification techniques were applied for the two extreme events. Because of
the differences in their features, we used the *Filtering method*, the *Relative Operating
Characteristics* (ROC) and *Area Under the ROC curve* and the *Taylor diagrams* for IOP13,
and the *Whisker diagrams* and the *Probability Distribution of Cyclone Center Occurrence* for
Qendresa. The results are described in the following subsections.

### 8.1. Statistical analysis: IOP13 Episode

Because IOP13 was a heavy rainfall episode, to quantitatively assess the impact on the short-
range forecasts from assimilating both *in-situ* conventional and reflectivity observations from
Doppler weather radars using the 3DVar and the EnKF DA algorithms, the accumulated
precipitation field will be used here.

### 8.1.1. Filtering Method

The FSS associated with the accumulated precipitation field is computed independently for the
three sub-regions R1, R2 and R3 highlighted in Fig. 7b, where the density of observation was
higher, using as threshold 1 mm·h$^{-1}$. In general, the comparison in terms of FSS (Fig. 8 a-c)
shows that EnKF outperforms 3DVar during the first 7 hours of free forecast in the three sub-
regions. As it was expected, the CNTRL experiments for both the EnKF and 3DVar outperform
the SYN experiments, where reflectivity observations were not considered. Moreover, Fig. 8a
shows that the 3DVar-CNTRL provides the worst scores, except for the first few hours of
simulation where 3DVar-CNTRL performs better than 3DVar-SYN. This is because the
information ingested from the radar using the 3DVar in that region is lasting no longer than 2
hours. Something similar happens with the EnKF after 4 hours. These results would agree with
past studies, showing similar behaviors (Carrió et al., 2016; Carrió et al., 2019).


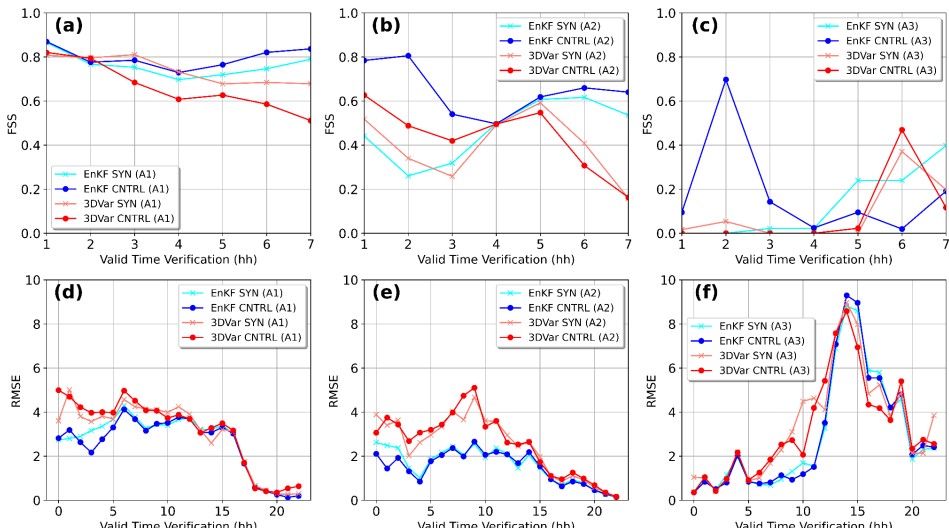

Figure 8. Upper panels: Evolution of the FSS during the first 7 hours of free forecast in the Italian sub-regions (a)
R1, (b) R2 and (c) R3, using a threshold > 1 mm·h⁻¹. Lower panels: Evolution of the RMSE during the first 24
hours of free forecast in the sub-regions (d) R1, (e) R2 and (f) R3. Simulations assimilating both conventional and
radar observations (CNTRL) and simulations assimilating only conventional observations (SYN) associated with
the 3DVar and the EnKF are shown here.

In addition to the FSS, we also compute the typical and widely used root-mean-squared-error
(RMSE) on the precipitation field for the first 24 hours for both EnKF and 3DVar simulations.
In general, the EnKF provides the lowest (best) RMSE scores, with respect to 3DVar. Also,
note that the impact of the assimilation of reflectivity observations does not last more than 4-6
hours, in accordance with past studies.
**8.1.2. ROC and AUC**
To strengthen how skillful are the different simulations performed by the 3DVar and the EnKF,
the *Receiver Operating Characteristic* (ROC) curve is used. The probability of exceeding a
given threshold is computed and verified against dichotomous observations. The ROC curve is
computed as follows: the model variable is interpolated to the observation locations and if the
model variable exceeds a given threshold, that model grid point is assigned a value of 1. On
the contrary, if the model value does not exceed that threshold, the assigned value is 0. The
same method is applied for the observations. Then, using these dichotomous values, the Hit
Rate and False Alarm scores are computed. This process is repeated, varying the threshold
value. Gathering the Hit Rate and False Alarm scores for the different thresholds, we obtain
the ROC curve. For the 3DVar, we get the Hit Rate and False Alarm scores by simply
interpolating the model values to the observation locations and apply the threshold criteria
explained above. In the case of the EnKF, the ensemble mean is used as the field to be
interpolated to the observation locations. The area under the ROC curve (AUC), which
measures the ability of the system to discriminate between the occurrence or nonoccurence of
the event, is also computed.
For the sake of brevity and because the results from the three sub-regions are similar, the ROC
and the area under the ROC curve are computed, accounting for all the observations within the



inner numerical domain. Specifically, to compute the ROC curves, we use the 3-hour (from 00
UTC - 03 UTC 15 Oct) and 6-hour (from 00 UTC - 06 UTC 15 Oct) accumulated precipitation
fields from the numerical model and the observed values registered by the rain gauges, using 1
mm and 10 mm as thresholds (Fig. 9).
Results show that EnKF clearly outperforms 3DVar for the different accumulated precipitation
rates and thresholds, depicting larger values of AUCs. An even bigger improvement is obtained
using a larger threshold (i.e., bottom row of Fig. 9) for EnKF, where the benefits of assimilating
radar observations are noticeable, in comparison with 3DVar. To better understand this result,
we inspected in more detail the 1-h and 6-h accumulated precipitation fields obtained from the
EnKF (CNTRL) and the 3DVar (CNTRL) and we compared those fields against the
corresponding observations (see Fig. A1 in the Appendix). The 1-h accumulated precipitation
(first row, Fig. A1) shows that the EnKF is localizing with high accuracy the regions where the
most intense precipitation was observed, that is near Tuscany and northern Italy. Also, 3DVar
correctly reproduces the rainfall in the regions affected by observed precipitation, although the
maximum amounts are centered over Liguria, instead of near Tuscany. In addition, the 3DVar
is also showing a tongue area of weak precipitation from Liguria to northern Italy, that does
not fit with the observations. Hence, although there are some differences between 3DVar and
EnKF for the 1-h accumulated precipitation field, because the accumulated precipitation values
are small, the ROC verification scores from the EnKF and 3DVar do not differ significantly.
However, in the case of the 6-h accumulated precipitation (second row, Fig. A1), the 3DVar
produces higher values of accumulated precipitation near Liguria, Tuscany and northern Italy
than the observed ones. Moreover, 3DVar is also misplacing the locations of the precipitation
for some places. On the contrary, the EnKF can (a) locate with enough accuracy the regions
where the accumulated precipitation was actually observed, (b) properly estimate the observed
intensity and (c) avoid spatial errors associated with the location where the precipitation was
produced. This is why ROC for the 6-hour accumulated precipitation obtained from the EnKF
produced a much better score than the 3DVar. We hypothesize that this difference could be
associated with the *static/climatological* background error covariance matrix used by the
3DVar. Because of the fast changes in the flow associated with the IOP13 case, using a
climatological background error covariance could not be as good as using a flow-dependent
background error covariance matrix, which is used in the EnKF.

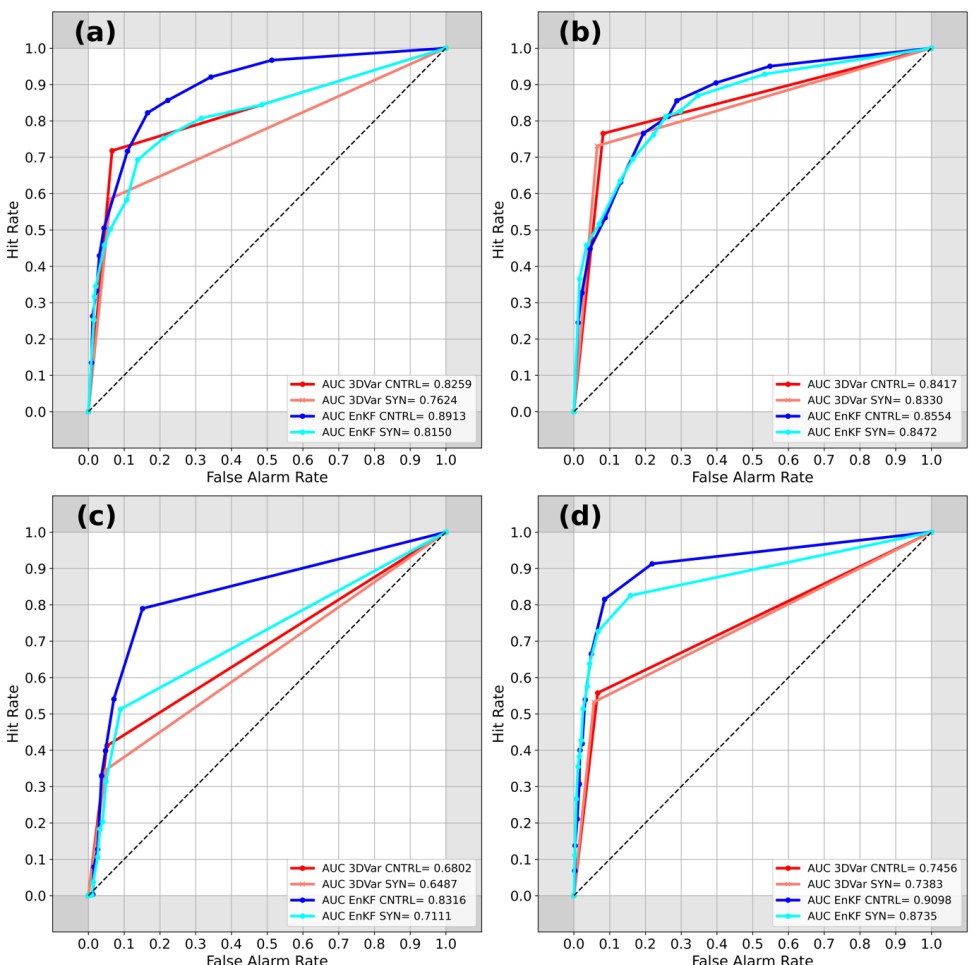

Figure 9. ROC curves and AUC associated with the 3DVar (red colors) and EnKF (blue colors) for the 3-hour
accumulated precipitation using (a) 1 mm and (b) 10 mm threshold and 6-hour accumulated precipitation using
(c) 1 mm and (d) 10 mm threshold, computed over the entire inner domain.

**8.1.3. Taylor Diagrams**
To strengthen the comparison of the DA schemes, the Taylor Diagram is used. This tool
provides us with extra information about the skill of each ensemble member in the case of the
EnKF. Here, we compute the Taylor diagram over the 6-hour accumulated precipitation field,
which is the range where the observations assimilated have more impact on the forecast.
Results show that the 3DVar and the ensemble mean of the EnKF provide similar results, with
similar correlations (0.50-0.61), similar root mean squared error and standard deviation that are
distributed symmetrically about the observation value, with the 3DVar overestimating the
standard deviation and the EnKF underestimating it (Fig. 10). However, if we consider each
ensemble member, we can observe that there is a cluster of the ensemble members of the EnKF
that provide better scores than the 3DVar. Although the difference between the EnKF and the
3DVar in this case is small, we can point out that the EnKF provides additional information
from their individual ensemble members. For instance, the individual ensemble members
showing higher correlation and standard deviation similar to the observations for this study are
the ones using Grell-Freitas cumulus parameterization in combination with the Yonsei
University planetary boundary layer scheme. Ensemble members associated with the lower
scores are those using Kain-Fritsch for the cumulus parameterization and the Mellor-Yamada-
Janjic for the planetary boundary layer scheme.

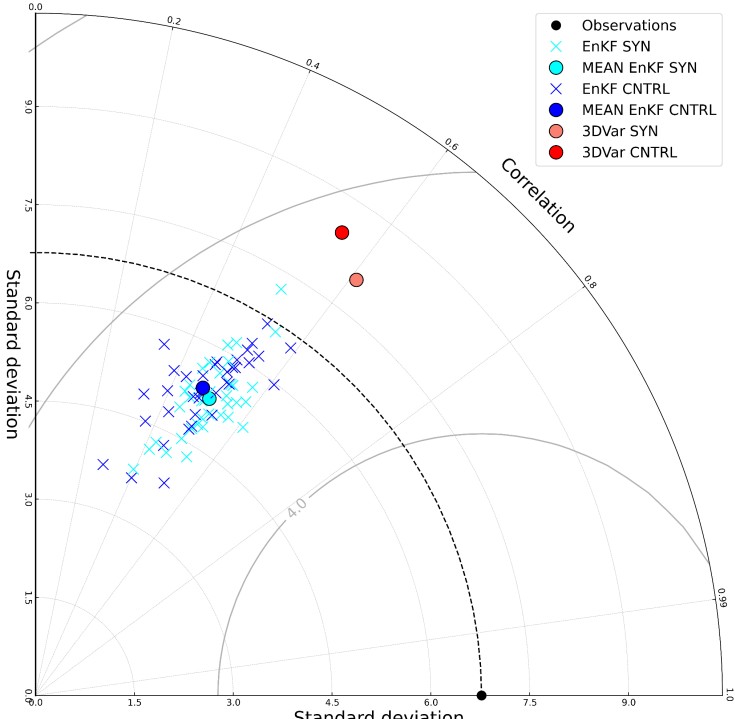


Figure 10. Taylor diagram performed by the 3DVar (reds) and EnKF (blues) for the 6-hour accumulated
precipitation valid at 06 UTC 15 October 2012.

## 8.2. Statistical analysis: Qendresa event

Typically, two key factors are investigated for Tropical cyclone forecasts: (a) the intensity and
(b) the trajectory followed by the cyclone. Therefore, to assess the impact of assimilating both
*in-situ* conventional and remote RSAMV observations using the 3DVar and the EnKF, these
two factors are considered.






### 8.2.1. Whisker Diagrams

For this event, the lack of *in-situ* observations over maritime regions poses a main challenge to properly verify the triggering and intensification of cyclones. Fortunately, the Qendresa medicane crossed just over Malta island, where a pressure drop greater than 20 hPa in 6 h, was registered by METARs at Malta airport, reaching a minimum of surface pressure of 985 hPa. Therefore, this METAR is used to quantitatively assess the skill of the different DA simulations. To compare the surface pressure registered at Malta with the different simulations, the full cyclone trajectory is used, and the grid point closest to Malta airport is selected. Finally, the surface pressure time series associated with that model grid point is compared with the values registered at Malta airport. Specifically, the surface pressure time series measured by METAR is compared with the different DA simulations from 3DVar and EnKF, such as the 3DVar_SYN, 3DVar_CNTRL, EnKF_SYN, and the EnKF_CNTRL (Fig. 11).

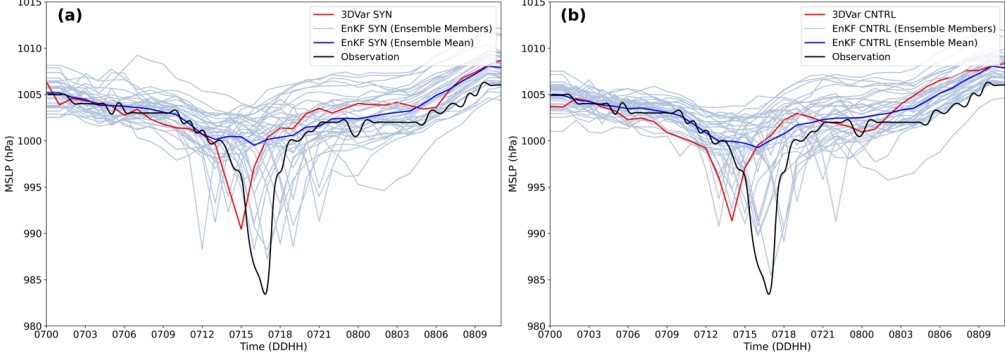

Figure 11. Temporal surface pressure evolution at the closes grid point to Malta for the (a) SYN and (b) CNTRL experiments associated with the EnKF (blue lines) and 3DVar (red lines), compared to the observed surface pressure registered by METARs in Malta's airport (black line).

Results from the assimilation of *in-situ* conventional observations show that the ensemble mean of the EnKF_SYN accurately fits the observations during the first hours of the forecast, from 00 UTC to 13 UTC 7 November (Fig. 11a), performing slightly better than 3DVAR_SYN. However, during the intensification phase, the ensemble mean of the EnKF_SYN barely shows the intensification of Qendresa, reaching minimum MSLP values of 1002 hPa. On the contrary, the 3DVar_SYN simulation depicts the intensification of the medicane, by deepening the MSLP and reaching values of 992 hPa, although a time shift of 3 hours is found (i.e., 15 UTC 7 November) (Fig. 11a). Finally, during the dissipation phase of Qendresa, the ensemble mean of EnKF_SYN is performing a bit better than the 3DVar_SYN (Fig. 11a). This interesting result clearly shows a limitation of the EnKF when applied to low-predictable weather events, such as Qendresa. The low predictability and the high sensitivity to the different physical parameterization schemes used for the forecast of this kind of event, lead to a very different behavior of each ensemble member. Consequently, some members could completely fail in the prediction of the weather event. In this situation, our small-to-moderate ensemble will probably produce a poor flow-dependent background error covariance matrix, which is key in DA, resulting in an analysis ensemble with large spread, for which ensemble mean will be smoothed out significantly. On the other hand, in such situations, we could think of using a climatological/static background error covariance matrix, as the one used in the 3DVar. If this


climatological background error covariance matrix is obtained with a large enough statistical
sample, it could produce much better results than using the flow-dependent background error
covariance computed with ensemble members that are not accurate enough, as we see in Fig.
11a when we compared the 3DVar (red line) with the EnKF ensemble mean analysis (blue
line). Also, it is important to note that although the ensemble mean of the EnKF_SYN is not
correctly reproducing the intensification of Qendresa, some of the ensemble members
accurately reproduce the observed MSLP both in deepening and timing. This suggests that
using an ensemble system, even having the above-mentioned problems, is still more useful than
using only a fully deterministic system such as the 3DVar, which cannot provide information
about the uncertainties of the system. Therefore, we can speculate that for extreme weather
events with low numerical predictability, a better approach could be using a Hybrid error
covariance model, where the forecast error covariance matrix is obtained linearly combining
ensemble-based covariance with static climatological error covariances (Hamill and Snyder
(2000); Lorenc (2003); Clayton et al., 2013; Carrió et al., 2021). The impact of using hybrid
DA to improve this kind of small-scale extreme weather events could be of great interest in the
weather forecast community, although it is beyond the scope of this study. For this reason, the
authors leave as future work the benefits of using hybrid error covariance models to improve
the forecast of extreme weather events in the Mediterranean basin.
Then, we evaluated the impact of assimilating both *in-situ* conventional and RSAMV
observations in the improvement of Qendresa intensity forecast. In this case, the results show
large similarities with the assimilation of only *in-situ* observations (Fig. 11b). In terms of the
3DVar, the MSLP signature is basically the same, without showing a clear signal of
improvement or diminishing, suggesting that the assimilation of RSAMVs is not enough to
significantly improve the low level relevant dynamical structures associated with the genesis
and intensification of Qendresa. However, in terms of the EnKF a clear improvement for a few
members is found, even if it is not affecting the mean value. Indeed, some of the ensemble
members depicting an intense cyclone far from the time when it was observed (approx. at 18
UTC 7 November), were corrected reducing spurious cyclones and the deepening of at least
one ensemble member close to the observed value (Fig. 11b). It can be observed that in the
EnKF_CNTRL, there are more ensemble members depicting a deep cyclone at the observed
time than in the case of the EnKF_SYN, showing the benefits of assimilating RSAMVs to
improve the intensification estimation of Qendresa.
To quantitatively assess the performance of the different DA experiments, we use the *lagged*
*correlation* technique computed between the model MSLP signatures and the observations.
This technique allows us to measure how the shape of the surface pressure evolution obtained
from the different simulations fits the shape of the observed MSLP, taking also into account
temporal shifting. The correlation is computed for the deterministic 3DVar, and for each
ensemble member from the EnKF. These results are shown using Whisker plots (Fig. 12).
Notice that a correlation of one means that the specific model field has the same 'V' pressure
shape evolution as the observation, and that the minimum for both is found at the same time.
For the 3DVar_SYN, the correlation is maximum and approximately equal to one when 1-hour
delay is applied to forecasts (Fig. 12a). Whiskers from EnKF_SYN show that none of the
ensemble members overcomes the maximum correlation value found in 3DVar_SYN.
However, when the assimilation of RSAMVs is added to the *in-situ* conventional observations,
it is found that the maximum correlation value associated with 3DVar_CNTRL using 2h of
delay applied to the forecasts, is surpassed by some of the ensemble members of the
EnKF_CNTRL, when a 3 or 4 hour of delay is applied (Fig. 12b).

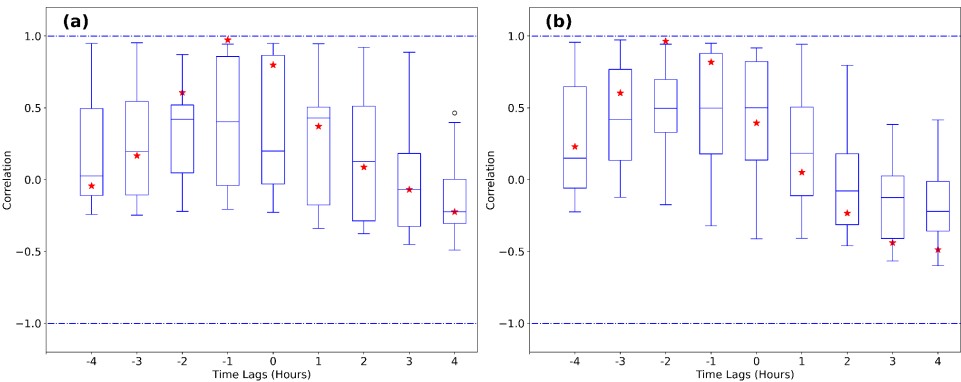

Figure 12. Whisker plots depicting the lagged correlation values between the observations and the EnKF (blue
boxes) and the 3DVar (red stars) for the (a) SYN and (b) CNTRL experiments. The correlation is computed
considering that the observed V-shape pressure signature associated with the observations is shifted 4 hours to the
left and 4 hours to the right.

**8.2.2. Probability Distribution of Cyclone Center Occurrence**
Due to the difficulty to accurately predict the observed trajectory of Qendresa (Pytharoulis et
al., 2018), the impact of assimilating different kinds of observations on the trajectory of the
medicane is investigated.
The 3DVar_SYN is capturing with enough accuracy the track of Qendresa during the first
hours (Fig. 13b). However, for 3DVar_SYN the trajectory of Qendresa leaving Malta diverges
from the observed trajectory, moving north-eastwards without showing the track-loop signal
observed by satellite imagery. To quantify the benefits of assimilating *in-situ* conventional
observations using the 3DVar or the EnKF, the probability of occurrence of a cyclone following
the track observed via satellite imagery is computed. For instance, we can see that the
probability of cyclone occurrence eastwards Sicily, where Qendresa made landfall while it was
doing a loop, is too small according to 3DVar_SYN (Fig. 13b). On the other hand, some of the
ensemble members depict a cyclone trajectory for EnKF_SYN that is largely shifted
southward, whereas some of them reproduce the loop trajectory that deterministic numerical
weather models miss performing (Fig. 13a). In addition, the probability of Qendresa occurrence
eastwards Sicily, is in this case larger than for 3DVar_SYN, showing the benefits of using the
EnKF against the 3DVar (Fig. 13a). Moreover, the EnKF_SYN ensemble trajectories, in
general, follow a 'V' shape (i.e., first moving towards the southeast, then moving to the east
and finally moving towards the northeast) similar to the trajectory observed via satellite
imagery. Although the shape of most of the EnKF_SYN trajectories agree with the
observations, the location is not accurate, showing a general shift towards the southeast.
If both *in-situ* conventional and RSAMV observations are assimilated, some of the ensemble
members from the EnKF_CNTRL shows more accurate trajectories in comparison with
EnKF_SYN: the loop trajectory is close to the observed region of eastern Sicily (Fig. 13c). An
improvement of the 3DVar_CNTRL trajectory by increasing the probability of cyclone
occurrence following the observed track is observed, especially eastern of Sicily. However,
3DVar experiments are not able to reproduce the looping trajectory observed via satellite
imagery (Fig. 13b-d). Hence, EnKF outperforms 3DVar showing some of the ensemble


members depicting a loop trajectory, although shifted southwards and producing a probability
of cyclone occurrence smaller than the 3DVAR ones.
Both the EnKF and the 3DVar still have difficulties in depicting accurately the track observed
by Qendresa, even after the assimilation of *in-situ* conventional and RSAMV observations.
Because RSAMVs are more useful in describing dynamical features on the upper levels of the
atmosphere, we hypothesize that ingesting them via DA may not be enough to correct key low-
level dynamical features. In this case, the assimilation of surface wind observations may help
to even improve these results. However, this is beyond the scope of this study and the authors
leave this question as future work, where other sources of information from satellites will be
assimilated to improve low-level thermodynamic aspects of extreme weather events, such as
medicanes.

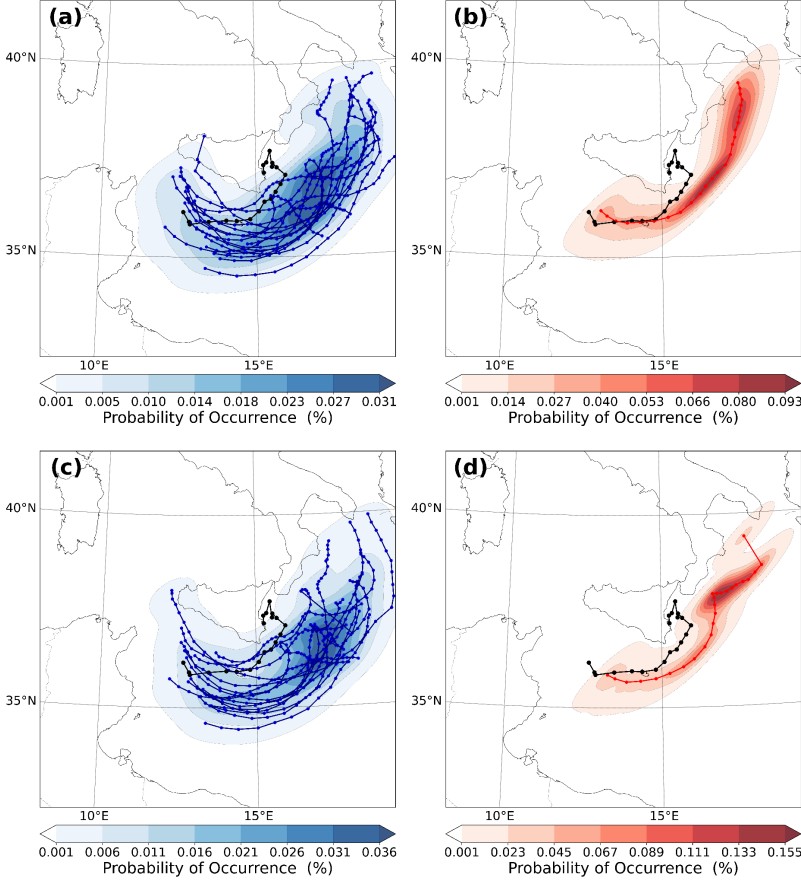

Figure 13. Probability of cyclone center occurrence computed using Gaussian KDE for (a) EnKF (SYN), (b)
3DVar (SYN), (c) EnKF (CNTRL) and (d) 3DVar (CNTRL), from 11 UTC 7 November to 12 UTC 8 November
2014. Qendresa's trajectory observed via satellite imagery is depicted in black.






## 9. Summary and Conclusions

In this study, we quantitatively assess the impact of two high-resolution DA techniques. Here,
we focus on the impact of assimilating observations to improve warning lead times of extreme
weather events. While previous studies often assimilate observations during the mature stage
of a weather event, when it is fully developed and no time for action remains, here the
observations are assimilated hours before the mature stage of the convective system is reached,
during the pre-convective stage. This approach enhances the accuracy of the pre-convective
environment, thereby increasing the time available for reaction and preparedness. To
quantitatively evaluate their forecast skill in improving the predictability of maritime events,
two extreme weather events triggered over the sea affecting populated coastal regions are used.
Nowadays, these weather events represent a serious challenge for the numerical weather
prediction community in terms of their accurate predictability, due to their initialization over
the sea, which are regions with a lack of in-situ observations, and thus their initial conditions
are poorly estimated. Furthermore, these convective systems evolved towards complex terrain
regions, increasing the predictability challenges. These two extreme weather events are known
as (a) the high precipitation event registered during the 13[th] Intensive Observation Period
(IOP13) affecting the western, northern and central parts of Italy, and (b) the intense Tropical-
like Mediterranean Cyclone (medicane) known as Qendresa, that affected the islands of
Pantelleria, Lampedusa, Malta and Sicily.
The two DA methods compared in this study for IOP13 and Qendresa are the variational 3DVar
and the ensemble-based EnKF, which are currently used in operational National Weather
Services worldwide. For the two events, both DA methods are used, and the type and number
of assimilated observations changes depending on the data availability. For Qendresa, we
assimilated (a) hourly *in-situ* conventional observations and (b) wind speed and wind direction
profiles of the entire atmosphere (RSAMVs) derived from geostationary satellites every 20-
min, providing high spatial and temporal resolution observations covering the Central
Mediterranean Sea, where Qendresa initiated and evolved. On the other hand, for the IOP13,
we assimilated (a) hourly *in-situ* conventional observations and (b) 15-min 3D reflectivity
observations from two type-C Doppler Weather Radars.
Because of the different thermodynamic characteristics associated with Qendresa and IOP13,
a set of different verification metrics were used for each of these extreme weather events. The
*Filtering method* (FSS and RMSE), the ROC/AUC and the *Taylor diagram* were used to verify
the numerical simulations from 3DVar and EnKF associated with IOP13. In the case of
Qendresa, we used the *Whisker diagrams* and the *Probability Distribution of Cyclone Center*
*Occurrence* verification scores. For the IOP13 event, the *Filtering method* and the *Taylor*
*diagram* verification scores indicate that the skill performance of the 3DVar and the EnKF is
similar, although the EnKF slightly overcomes the 3DVar. In addition, it was observed that the
assimilation of spatial and temporal high-resolution reflectivity observations significantly
improved the forecast for both 3DVar and EnKF, showing the key role of this type of
observation. On the other hand, the ROC and AUC scores clearly show that EnKF outperforms
3DVar. For the Qendresa event, although the ensemble mean of EnKF provides the worst
results, in terms of the intensity of the medicane with respect to 3DVar, some of the EnKF
ensemble members provide better results than 3DVar. This result suggests how important it is
using an ensemble forecast system to predict extreme weather events at high spatial and
temporal resolution. In terms of the trajectory of the cyclone, it is also shown that using the
EnKF provides a more realistic insight of the real trajectory Qendresa followed.


Although the EnKF technique has shown in general better performance against the 3DVar for the two extreme weather events analyzed in this study, it is also important to account for the computational resources required to use them. In this sense, the 3DVar requires much less computational resources than the EnKF because it does not need to build an ensemble of considerable size, and it does not need either to simulate model trajectories between the assimilation of a set of observations at time $t_1$ and the subsequent set of observations valid at $t_2$. This makes the 3DVar appealing because it is much faster and cheaper than the EnKF, and it makes this technique particularly suitable for operational purposes at the small weather forecast centers.

Another interesting result that we have shown in this study is that depending on the level of predictability of the weather event and its sensitivity to numerical physical parameterizations used to build our ensemble, the 3DVar performs better than the EnKF ensemble mean. We speculated that this is linked to the way the background error covariances from these two methods are built. Based on this, we suppose that a better approach could be using Hybrid error covariance models, where the forecast error covariance matrix is obtained linearly combining the ensemble-based error covariance from the EnKF and the static climatological error covariance matrix from the 3DVar. Further work will investigate the impact of using hybrid DA schemes in comparison to use standard 3DVar or EnKF. As a case study, a catastrophic and deadly flash flood event affecting the Balearic Islands will be used to quantitatively assess the skill performance of the hybrid DA scheme against the EnKF and a more advanced version of the 3DVar, which is known as the 4DVar. In this case, most of the ensemble members of the EnKF did not reproduce the convective cells that later resulted in the flash flood episode. This is a key problem in current ensemble-based DA research. In this scenario, it is expected that the hybrid error covariance matrix will be more precise than the one derived from the ensemble members or from climatology, which on their own are not properly reproducing key aspects of this extreme weather episode. High temporal and spatial observations from Doppler Weather radars, such as reflectivity and radial wind velocities, will be assimilated for this case to obtain accurate analysis and thus, improve the short-range forecast of this catastrophic flash-flood event.

**Acknowledgements:**

The first author acknowledges the *Ministerio de Universidades (Plan de Recuperación, Transformación y Resiliencia)* funded by the European Union (NextGenerationEU), with the participation of the University of the Balearic Islands, which supports his current postdoctoral position as a *Maria Zambrano* Fellowship. However, it should be noted that the viewpoints and opinions expressed in the relevant cases, or resulting from the grant, are the sole responsibility of the author or authors and do not necessarily represent the views of the European Union or the European Commission. Neither the European Union nor the European Commission can be held liable for any of the opinions expressed. This research is also sponsored by the Ministerio de Ciencia e Innovación-Agencia Estatal de Investigación TRAMPAS (PID2020-113036RB-I00/AEI/10.13039/501100011033). The authors thankfully acknowledge Météo-France for supplying the data and HyMeX database teams (ESPRI/IPSL and SEDOO/OMP) for their help in accessing the data. The author also acknowledges the computer resources at MareNostrum IV and CRAY supercomputers, as well as the technical support provided by the Barcelona Supercomputer Center (RES-AECT-2017-1-0014, RES-



AECT-2017-2-0014) and ECMWF data center, required to perform the high-resolution
simulations presented in this study.

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

**Appendix**

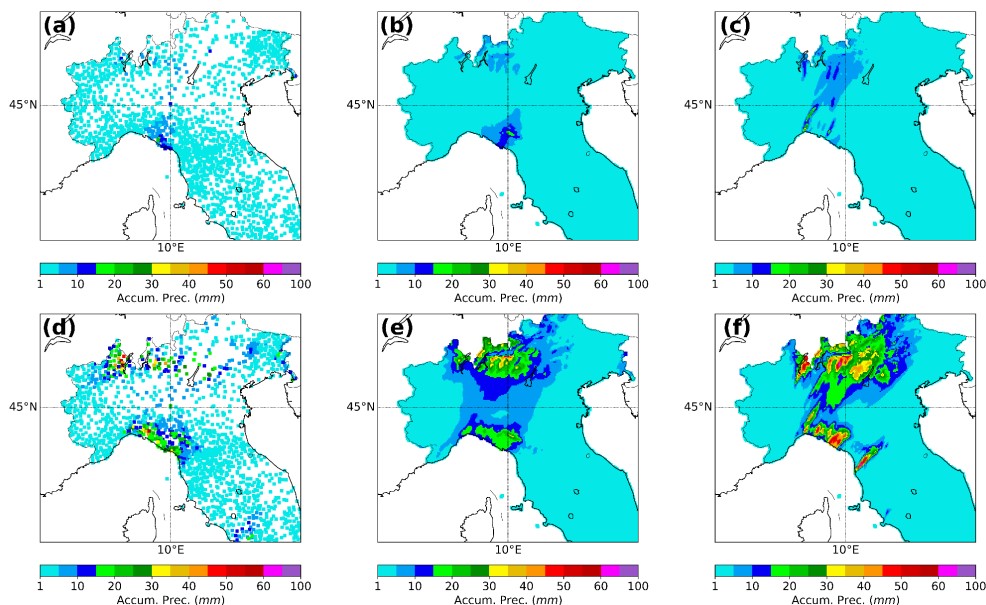

Fig. A1. 1-h accumulated precipitation computed from 00-01 UTC 15 October 2012 associated with (a)
Observations, (b) EnKF (CNTRL) and (c) 3DVar (CNTRL). 6-h accumulated precipitation computed from 00-
06 UTC 15 October 2012 associated with (d) Observations, (e) EnKF (CNTRL), (f) 3DVar (CNTRL).
**Author Contribution**
**D. S. Carrió:** Conceptualization, Methodology, Software, Validation, Formal analysis,
investigation, writing-original draft, writing-review & editing, visualization, supervision; **V.**
**Mazzarella:** formal analysis, writing-review; **R. Ferretti**: formal analysis, writing-review,
supervision.
**Competing Interests**
The authors declare that they have no conflict of interest.