# Peer review of "High-Resolution Data Assimilation for Two Maritime Extreme Weather 1 Events: A comparison between 3DVar and EnKF. 2 3 4 Diego S. Carrió1, Vincenzo Mazzarella2, Rossella Ferretti2 5 6 1Meteorology Group, Department of Physics, University of the Baleari"

_Natural Hazards and Earth System Sciences, 2024_

## Author Comment (AC1)

**Reviewer #1:**

**General Comments:**

 1. *Absence of NODA in comparisons/figures*

*Although announced in section 6.3, the NODA is not part of any figure as far as I can see. Therefore, the reader can't be sure that the assimilation improves forecasts. It could be that both methods are not much different from NODA.*

We thank the reviewer for this important observation. We agree that including the NODA results in the comparisons is essential to properly assess the impact of the assimilation process.

In response to this comment, we have now incorporated NODA results into the relevant figures and discussions. The updated figures are included in the Section "New Figures" below in the present document, ensuring that the impact of data assimilation is clearly demonstrated by comparing EnKF, 3DVar and NODA runs. Additionally, we have revised the corresponding discussion to explicitly analyze how each DA method performs relative to the NODA run.

We appreciate the reviewer's input, as this addition strengthens the conclusions regarding the effectiveness of data assimilation in improving forecasts.

*2) Inconsistency of verification measures (shown in results) and indicated aims (section 2)*

*The paper intends to show ... (indicated aims, L161ff)*

 1. *the improved prediction of small-scale extreme weather events*
 2. *enhanced accuracy of atmospheric conditions in the pre-convective environment*
 3. *impact of assimilating in-situ conventional and remote sensing observations*

*Ad 1: There is no comparison to NODA, thus no improvement measurable. Moreover, the paper evaluates precipitation FSS(>1mm/h), RMSE(1h/6h), but does show to which extent the observed extreme precipitation could be forecasted.*

*Ad 2: Where did you evaluate the pre-convective environment?*

We thank the reviewer for this valuable comment. We acknowledge the initial concerns regarding the inconsistency between the verification measures and the stated aims of the study. However, these issues have already been addressed in response to the other reviewer's comments, and we have made significant revisions to improve the clarity of the study's objectives. Consequently, the aims outlined in the original manuscript have been updated in the revised version. Regarding Ad1, we have now incorporated NODA results into the relevant figures and discussions, ensuring that the impact of data assimilation is clearly demonstrated. These additions allow for a proper evaluation of the extent to which both 3DVar and EnKF improve forecasts relative to a no-data-assimilation scenario. We have also revised the precipitation verification by incorporating higher thresholds for FSS and ensuring that FSS and RMSE calculations are based on the same time window. This provides a clearer assessment of the model's ability to capture intense precipitation events, making the results more relevant to the study's stated objectives. In terms of Ad 2, we agree that our aim (2) in L165 was not clearly stated

and may have been misleading. Most DA studies focus on assimilating observations once the weather phenomena has already initiated, rather than incorporating observations several hours before the event begins (i.e., in the pre-convective phase). To the best of our knowledge, very few studies focus explicitly on DA strategies in pre-convective conditions, particularly in the Mediterranean region. This distinction underscores the novelty of the present study and its contribution to the existing body of research. This study aims not to analyze how the pre-convective environment is modified after assimilation, but rather to evaluate the forecast impact of assimilating observations during the pre-convective phase, as opposed to assimilating observations after convection has initiated.Our objective is to assess whether assimilating pre-convective observations ultimately leads to an improved forecast of extreme weather events. The improvements observed in the forecasted convective evolution and precipitation fields suggest that assimilating data in the pre-convective phase contributes to a better representation of the event.

To ensure clarity, we have revised the objectives of the study, which now reads as follows:

"*On overall, this study aims at:*

*(a) Assessing the impact of 3DVar in comparison with the EnKF system to predict small-scale extreme weather events initiated over maritime regions with lack of in-situ observations.*

*(b) Investigate the potential of using 3DVar and EnKF in the pre-convective environment, hours before the mature stage of convective systems are reached, to improve forecast lead time and warning capabilities for extreme weather events.*

*(c) Compare the forecast impact from assimilating in-situ conventional observations in comparison to assimilating high spatial and temporal resolution data from remote sensing instruments.*

*(d) Provide a quantitative assessment between two different DA schemes using several statistical verification methods.*"

We appreciate the reviewer's feedback, as it has helped us refine the presentation of our study's objectives.

*3) Inconsistency of conclusions and results*

*L974-975: "Similar skill" of EnKF and 3DVar in FSS (Fig8) and Taylor diagram (Fig10)*

*That contradicts what I see in Fig 8 and 10.*

We thank the reviewer for this observation. We agree that this statement does not accurately reflect the results shown in these figures.

In response to this comment, we have modified this sentence to ensure consistency between the results and the discussion.

This sentence now reads as follows:
"*For IOP13, both the Filtering method and Taylor diagram verification show that EnKF slightly outperforms 3DVar, though the differences are not substantial.*

We appreciate the reviewer's feedback, as it has helped improve the clarity and accuracy of our conclusions.

*L976: "significantly improved the forecast". There was no comparison to NODA. Thus no improvement visible.*

We thank the reviewer for this comment. We agree that the statement in L976 regarding "significantly improved the forecast" was initially made without a direct comparison to NODA, making it difficult to quantify the improvement.

This issue has already been addressed in response to previous comments, as we have now included the NODA experiment in our analysis. The updated figures and discussions clearly demonstrate the impact of data assimilation relative to a no-data-assimilation experiment, providing a more robust assessment of forecast improvements.

We appreciate the reviewer's input, as it has helped ensure that our conclusions are better supported by the presented results.

*L979: "EnKF provides worst results." This is not a disadvantage of EnKF. The ensemble mean of a cyclone pressure field is as useful as the ensemble mean of a precipitation field. As it is not Gaussian, it should not be expected to perform well.*

We thank the reviewer for this insightful comment. We agree that this is not a disadvantage of EnKF, but rather an inherent characteristic of ensemble-based methods. While the ensemble mean provides valuable information, it is not always the most appropriate metric for variables with non-Gaussian distributions, such as cyclone pressure fields or precipitation fields. In such cases, ensemble forecasts are better interpreted in a probabilistic framework, rather than relying solely on the ensemble mean.

In response to this comment, we have revised the discussion to more accurately reflect these considerations and removed the misleading statement. Now this sentence reads as follows:

"*For the Qendresa event, while the **ensemble mean of EnKF underestimates the intensity of the medicane compared to 3DVar**, some individual **EnKF ensemble members produce more accurate results than 3DVar**.*"

We appreciate the reviewer's feedback, as it has helped us improve the interpretation of our results.

**Minor comments:**

*Figure 8: The authors employ FSS of precipitation >1 mm/h for three regions (Fig 8a-c). This score evaluates correct positioning of precipitation in forecasts, but doesn't show improved prediction of extreme events (1 mm/h is hardly extreme).*

We thank the reviewer for this comment. This concern was also raised by Reviewer 1, and we have already addressed it by incorporating higher precipitation thresholds in the FSS analysis to better assess the prediction of extreme precipitation events. We have updated Fig. 8 to include the new FSS results using progressively higher thresholds, ensuring a more meaningful evaluation of extreme

precipitation forecasts. The updated figures are included in the Section below "**New Figures**", and the corresponding discussion has been modified accordingly.

We appreciate the reviewer's input, as this improvement enhances the robustness of our precipitation verification.

*Figure 8: It is unclear what RMSE shows. Is it RMSE of ensemble mean prediction of precipitation? [mm/h]?*

The RMSE shown in Figure 8 corresponds to the root mean square error of the predicted precipitation field, evaluated against observations not assimilated. Specifically, for the 3DVar, RMSE is computed from the deterministic forecast, and for EnKF, it is computed from the ensemble mean precipitation field. The RMSE values are expressed in mm/h.

To improve clarity, we have updated the figure caption and revised the text in the manuscript to explicitly state this information. We appreciate the reviewer's suggestion, as it helps ensure that the methodology is clearly communicated.

*Figure 8: Is the FSS of EnKF computed from the ensemble mean forecast or from the whole ensemble?*

We thank the reviewer for this comment. The FSS for EnKF is Figure 8 has been computed from the ensemble mean forecast, rather than from individual ensemble members.

To ensure clarity, we have updated the figure caption and revised the corresponding section in the manuscript to explicitly state this.

*Figure 10: Which observations have been used for this figure?*

The observations used in Figure 10 refers to the precipitation observations, as those used in Figure 8.

To improve clarity, we have updated the figure caption and revised the manuscript text to explicitly state this.

*The paper is 31 pages. I suggest to shorten the text in the interest of the reader and journal guidelines. For example, the introduction is very long and does not always on point (why are particle filters discussed?). The methods contain a revision of DA equations. I don't see how that serves the rest of the paper.*

We thank the reviewer for this suggestion. We agree that the manuscript can be shortened to improve readability and better align with journal guidelines. In response to this comment, **we have reduced the section introducing EnKF and 3DVar, keeping only the essential details relevant to this study**. While we have removed the general revision of DA equations, we have retained key information on the specific parameters used to configure these DA schemes, as this was explicitly requested by reviewers in previous studies.

*L162-172: These are general points and considering the limited set of observations for verification, such general questions cannot reasonably be answered, as you state in L174-177. Maybe you can edit L161 to include something like "we address these questions for two high-impact cases"*

We thank the reviewer for this comment. The section L161-172 has been revised following suggestions from the other reviewer, and the concerns raised here are no longer applicable. Specifically, we have clarified the study's objectives, ensuring that they align with the limited set of observations and case study approach.

We appreciate the reviewer's feedback, as it has contributed to improving the focus and clarity of our manuscript.

*L162, and others: "high-resolution data assimilation": I don't see what the difference between "high-resolution 3DVar" and 3DVar is. Clearly, if applied at high resolution, any method becomes a high-resolution method. Is there any more to it?*

We thank the reviewer for this comment and appreciate the opportunity to clarify our use of the term "high-resolution data assimilation".

In this study, "high-resolution" refers not only to the spatial resolution of the model but also to the high-temporal frequency of data assimilation cycles. This approach significantly increases computational demands, as it requires assimilating observations at short intervals while maintaining fine-grid resolutions.

While 3DVar and EnKF can be applied at various resolutions, the combination of high spatial and temporal resolution DA is computationally expensive and less commonly explored in the literature. Our study aims to evaluate these methods under this computationally demanding configuration, which differs from many conventional DA studies that use lower temporal update frequencies or coarser spatial resolutions.

To clarify this point, we have revised the manuscript to better define what we mean by high-resolution data assimilation in this context. We appreciate the reviewer's input, as it helps improve the precision of our terminology.

*L171-172: Isn't this the same as point (a) before?*

We thank the reviewer for this comment. This section has been revised following suggestions from the other reviewer, and the concerns raised here are no longer applicable because this section is now different. The structure and wording have been adjusted to improve clarity and avoid redundancy.

*Missing table: It would be useful to collect the assimilated observation types per case in a table.*

We thank the reviewer for this helpful suggestion. We agree that including a table summarizing the assimilated observation types for each case will improve the clarity of the manuscript.

In response to this comment, we have added a new table that lists the types of observations assimilated in each case study. This table ensures a clearer presentation of the observational data used in the assimilation process.

| Event | Observation Type | Data Sources | Assimilation Frequency | Coverage | Additional Processing |
|-------|------------------|--------------|------------------------|----------|-----------------------|
| IOP13 | Conventional *in-situ* data | MADIS (NOAA) | Hourly | Entire Domain | Quality-controlled |
| IOP13 | Radar Reflectivity | Météo-France Doppler Weather Radars (Aleria & Nimes) | Every 15 minutes | Ligurian Sea & Gulf of Genoa | Quality controlled and Interpolated using Cressman Objective Analysis (6 km grid) |
| Qendresa | Conventional *in-situ* data | MADIS (NOAA) | Hourly | Mediterranean Region | Quality-controlled |
| Qendresa | Satellite-Derived Winds (RSAMVs) | EUMETSAT (SEVIRI instrument onboard MSG) | Every 20 minutes | Entire atmosphere over the Mediterranean Region | Quality-controlled, superobbing (128x128 km, 25 hPa vertical) |

*L940: high-resolution DA techniques*

*Well, plain-vanilla 3D-Var is not a high-resolution DA technique, especially without hydrometeor control variables.*

We thank the reviewer for this comment and appreciate the opportunity to clarify our terminology. We acknowledge that standard 3DVar, particularly without hydrometeor control variables, is not inherently a high-resolution DA technique.

In this study, we refer to "high-resolution DA" in the context of both high spatial and high temporal resolution, rather than implying modifications to the 3DVar formulation itself. Our experimental setup involves frequent assimilation cycles and fine-grid numerical simulations, which significantly increase the computational demands and are less commonly explored in previous studies.

To prevent any misunderstanding, we have revised the wording in the manuscript to ensure that our use of "high-resolution DA techniques" is accurately conveyed. We appreciate the reviewer's feedback, as it has helped improve the precision of our terminology.

*Figure 6: The figure is split over two pages. It would be good if it would not be separated from the caption. Labels a-f could be replaced by SYN, CNTRL, NODA, if possible.*

We thank the reviewer for this suggestion. We agree that keeping Figure 6 and its caption on the same page will improve readability and presentation. We have adjusted the formatting to ensure that the figure is not split across multiple pages, and we have also updated the labels (a-f) to more descriptive names such as SYN, CNTRL and NODA, making it easier for the reader to interpret the figure.
We appreciate the reviewer's input, as these improvements enhance the clarity and accessibility of the figure.

*L566: The year should be 2012 not 2021.*

We thank the reviewer for catching up this typo. We have corrected the year from 2021 to 2012 in L566 to ensure accuracy. We appreciate the reviewer's attention to detail.

*Finally, I would suggest to mention the opportunities from satellite data assimilation for convective-scale forecasting. Future studies could benefit greatly from that.*

We thank the reviewer for this valuable suggestion. We agree that satellite data assimilation presents significant opportunities for improving convective-scale forecasting, particularly in data-sparse maritime and remote regions.

In response to this comment, we have included the following in the conclusions section, highlighting the potential benefits of future studies incorporating satellite-based observations:

"*In addition, it is important to highlight that satellite-based data assimilation presents a significant opportunity for advancing convective-scale forecasting, particularly in data-sparse maritime regions such as the Mediterranean, where the formation of extreme weather events like tropical-like cyclones is increasingly impacting densely populated areas. Future studies integrating high-resolution satellite observations, such as cloud top highs, thermodynamic profiles or cloud properties, could further enhance the accuracy of convective-scale predictions, improving early warning capabilities and disaster preparedness.*"

We appreciate the reviewer's input, as this addition strengthens the discussion on potential future advancements in data assimilation techniques.

---

## Author Comment (AC2)

**Reviewer #2:**

**General Comments:**

1. *One of the main objectives of the paper is to compare the EnKF and 3DVAR approaches to determine which method provides better forecasts. However, the comparison is based on only two case studies, which is too limited a sample to adequately answer this question. If the paper were introducing a new method, using one or two example case studies might be sufficient. However, for a comparison of two existing methods, a larger number of cases is necessary to draw a robust conclusion.*

We sincerely thank the reviewer for this valuable observation. While we understand that a larger number of case studies would allow for a more statistically robust comparison of the two methods, we would like to clarify that the primary objective of this study is not to perform a statistical evaluation of the relative performance of EnKF and 3DVar over a wide range of cases. Instead, the goal of this paper is to compare the two techniques in two specific severe weather events in the Mediterranean region to highlight their respective strengths and limitations under contrasting conditions. This approach provides insights into their practical characteristics and implications, which we believe offer significant value to the research community.

It is worth noting that papers comparing different Data Assimilation (DA) methods using two – and even one – case studies are common in the literature. Some examples of papers employing a similar approach we have just used here are listed below:

**1.** Mazzarella, V. et al., 2017. Comparison between 3D-Var and 4D-Var data assimilation methods for the simulation of a heavy rainfall case in central Italy. *Advances in Science and Research*

**2.** Tiwari, G. et al., 2022. Predictive skill comparative assessment of WRF 4DVar and 3DVar data assimilation: An Indian Ocean tropical cyclone case study. *Atmospheric Research*

**3.** Sun, J. et al., 2013. Radar data assimilation with WRF 4D-Var. Part II: Comparison with 3D-Var for a squall line over the US Great Plains. *Monthly Weather Review*

**4.** Mazzarella, V. et al., 2020. Reflectivity and velocity radar data assimilation for two flash flood events in central Italy: A comparison between 3D and 4D variational methods. *Quarterly Journal of the Royal Meteorological Society*

We have further revised the manuscript to explicitly clarify the scope of this study in the introduction and conclusion sections to prevent potential misinterpretation. Specifically, the following sentences have been added:

- Introduction Section (page 4, Lines 149-180):

    *"Recent convective-scale DA studies have primarily focused on the mature stage of weather events (e.g., Wheatley et al., 2015; Jones et al., 2016; Yussouf et al., 2020). However, at this stage, the system is already well-developed and likely impacting the population, limiting the effectiveness of DA in terms of forecast lead time. In such cases, the potential for early warnings and mitigation actions is significantly reduced, as there is little time left to respond and minimize socio-economic impacts. Despite its potential benefits, very few studies have explored the role of DA in the pre-convective stage, where assimilating observations before convection initiates could significantly improve forecast lead time, providing advanced warnings and allowing decision-makers to act proactively. In this study, we compare the performance of two widely used DA techniques – 3DVar and EnKF – in enhancing the*

*forecast lead time for two extreme weather events initiated over the sea, a data-sparse region where observational constraints pose additional forecasting challenges, affecting populated coastal regions in the Mediterranean basin. It is important to emphasize that this study does not aim to derive statistically significant conclusions. Instead, the main objective is to compare the performance of EnKF and 3DVar in two distinct extreme weather events, each characterized by unique atmospheric conditions and observational limitations. The two extreme weather events selected for this study are: (a) the heavy rainfall episode, IOP13, affecting coastal regions of Italy during October 2012 (Pichelli et al., 2017) and (b) the low-predictable Mediterranean Tropical-like cyclone (medicane), Qendresa, affecting Sicily in November 2014 (Pytharoulis et al., 2017; Pytharoulis, 2018; Cioni et al., 2018; Di Muzio et al., 2019).*

*On overall, this study aims at:*

*(a) Assessing the impact of 3DVar in comparison with the EnKF system to predict small-scale extreme weather events initiated over maritime regions with lack of in-situ observations.*

*(b) Investigate the potential of using 3DVar and EnKF in the pre-convective environment, hours before the mature stage of convective systems are reached, to improve forecast lead time and warning capabilities for extreme weather events.*

*(c) Compare the forecast impact from assimilating in-situ conventional observations in comparison to assimilating high spatial and temporal resolution data from remote sensing instruments.*

*(d) Provide a quantitative assessment between the different DA schemes by means of using several statistical verification methods."*

- Conclusion Section (page 30, Lines 940-967):

"*This study provides a quantitatively assessment of the impact of two widely used DA techniques – 3DVar and EnKF – on the predictability of maritime extreme weather events. The focus is on evaluating their potential to improve forecast lead time by assimilating observations during the pre-convective stage, rather than the mature stage, when the event is already fully developed and there is limited time for preparedness and response. To evaluate the performance of 3DVar and EnKF in improving the predictability of maritime events, we analyze two high-impact weather events triggered over the sea that later affected densely populated coastal regions. These two extreme weather events are known as (a) the high precipitation event registered during the 13[th] Intensive Observation Period (IOP13) affecting the western, northern and central parts of Italy, and (b) the intense Tropical-like Mediterranean Cyclone (medicane) known as Qendresa, that affected the islands of Pantelleria, Lampedusa, Malta and Sicily. These weather events pose a serious challenge for the numerical weather prediction community in terms of their accurate predictability, due to their initialization over the sea, which are regions with a lack of in-situ observations, and thus their initial conditions are poorly estimated. Furthermore, their evolution over complex terrain regions introduces additional forecasting challenges.*

*For these two extreme weather events, both 3DVar and EnKF DA methods were applied, with the type and number of assimilated observations varying based on the data availability. For Qendresa, we assimilated (a) hourly in-situ conventional observations and (b) wind speed and wind direction profiles of the entire atmosphere (RSAMVs) derived from geostationary satellites every 20-min, providing high spatial and temporal resolution observations covering the Central Mediterranean Sea, where Qendresa initiated and evolved. On the other hand, for the IOP13, we assimilated (a) hourly in-situ conventional observations and (b) 15-min 3D reflectivity observations from two type-C Doppler Weather Radars.*"

We hope this clarification adequately addresses the reviewer's concern and provides a better contextual understanding of the study's objective.

2. *The comparison is made between the two methods but does not include the background NODA simulation. To effectively assess the added value of the two methods, especially considering that one is computationally more expensive, it is essential to include the NODA simulation in the comparison. This would allow for an evaluation of how much the simulations have improved over the background using the different data assimilation methods. Moreover, taking the first case, IOP13, as an example, a run starting on the 15th at 00:00 is used as the NODA simulation. While this may be acceptable for assessing improvement from an operational perspective, to evaluate the impact of DA, it would be better to include a simulation with the same initial and boundary conditions as the runs with DA. The same consideration applies to the second case as well.*

We thank the reviewer for this important comment. We agree that including the NODA simulation in the comparison is essential to effectively assess the added value of the two data assimilation methods, particularly given their differences in computational cost.

In response to this comment, we have now incorporated NODA results into the relevant figures and discussions. These additions ensure that the impact of 3DVar and EnKF can be quantified relative to a no-data-assimilation scenario, providing a clearer assessment of their benefits. We have included a NODA experiment initialized with the same conditions as the DA runs, ensuring a fairer comparison of forecast improvements due to data assimilation. This modification applies to both case studies (IOP13 and Qendresa).

We appreciate the reviewer's suggestion, as these additions enhance the robustness and validity of our analysis.

3. *For the EnKF, a multi-physics approach is used, while for the 3DVar, a deterministic run is adopted. Has it been evaluated whether the ensemble multi-physics approach provides an advantage for these cases compared to the deterministic run, regardless of the data assimilation? Or, at the very least, has the behavior of the EnKF simulation with the same physical configuration as that used for the 3DVar been analyzed? The paper does not address this consideration, making it impossible to separate the advantage of using a multi-physics ensemble approach from a deterministic run from the benefits of the two data assimilation methods, given that the multi-physics ensemble inherently offers numerous advantages.*

We thank the reviewer for raising this point and appreciate the opportunity to clarify this aspect of our study.

Firstly, it is important to acknowledge that the comparative analysis of ensemble multi-physics versus deterministic approaches is well-established in the literature and does not represent a novel aspect.
Secondly, regarding the reviewer's question about using the same physical configuration for the EnKF as the one employed for the 3DVar, we would like to emphasize the following. The EnKF method inherently relies on an ensemble-based approach to represent flow-dependent background error covariances, which are crucial for its proper functioning. The use of a multi-physics ensemble in this study is an integral part of the EnKF framework, as it enables the generation of diverse ensemble members and, consequently, a more accurate representation of system uncertainty. Conversely, 3DVar adopts a deterministic approach that assumes static, isotropic background error covariances, usually derived from a climatological background. These

distinctions in designs and purpose highlight the inherent differences between the two methods. Running the EnKF with the same deterministic configuration as 3DVar would not align with the principles of the EnKF and would likely lead to inaccurate results due to the lack of ensemble variability, which is essential for computing accurate flow-dependent error covariances.

We wish to emphasize that the main goal of this paper is to evaluate the performance of two widely used data assimilation strategies—EnKF and 3DVar—when applied to extreme weather phenomena in maritime regions, which are characterized by a lack of observational data. Typically, these data assimilation systems are tested over land regions where observational coverage is dense, presenting fewer challenges. However, their performance in the context of extreme weather events over maritime regions remains largely unexplored, to the best of our knowledge. While the questions raised by the reviewer are indeed relevant and have been extensively addressed in the existing literature, we believe that delving into these aspects within this study would not provide new or meaningful information to the community.

We hope this explanation clarifies the scope and objectives of our study and adequately addresses the reviewer's concern.

4. *In this work, operational evaluation is frequently discussed, but, for example, in the case of IOP13, the rainfall between 00 and 06 is analyzed from a simulation that starts at 00. In reality, this means that data assimilation occurs right up to the beginning of the event. In an operational context, to assimilate data up to 00:00 on the 15th, as shown in the scheme in Figure 6, it would be necessary to wait at least 15-20 minutes (or even more, it depends on the observations assimilated) after 00:00 on the 15th to acquire the latest observational data before starting the run. The 3DVar is relatively fast as an approach, especially if, as in this case, the observations assimilated are not very dense over the entire domain. But what time would the EnKF run be available? The risk with simulations structured in this way is that the forecast becomes available much later. In an operational context, which is repeatedly emphasized as a goal in the paper, evaluating the cumulative rainfall from 00 to 06 on the 15th effectively means validating rainfall that would not be forecasted but partially in hindcast. Actually, in Figure A1, the hourly precipitation between 00 and 01 is evaluated. Similar considerations can be made for the initial part of the second case, which, however, being evaluated over a longer period, is less affected by this issue.*

We thank the reviewer for this detailed comment, and we agree that the way we have explained this aspect in the paper may lead to misinterpretation. We appreciate the opportunity to rephrase and clarify this point.

The configuration used in this paper was designed to represent a real event, rather than being optimized for an operational setting, where computational constraints are not a limiting factor. This strategy is not novel and has been utilized in previous studies:

1. Carrió, D. S., & Homar, V. (2016). *Potential of sequential EnKF for the short-range prediction of a maritime severe weather event*. Atmospheric Research, 178, 426-444.

2. Carrió, D. S., Homar, V., & Wheatley, D. M. (2019). *Potential of an EnKF storm-scale data assimilation system over sparse observation regions with complex orography*. Atmospheric Research, 216, 186-206.

3. Carrió, D. S. (2023). *Improving the predictability of the Qendresa Medicane by the assimilation of conventional and atmospheric motion vector observations. Storm-scale analysis and short-range forecast*. Natural Hazards and Earth System Sciences, 23(2), 847-869.

While we acknowledge that real-world operational systems must address latency in data acquisition and computational speed, such operational constraints fall beyond the scope of this study. The primary goal of this paper is to provide scientific insights into the potential performance of EnKF and 3DVar under idealized conditions, rather than to propose practical operational guidelines.

To avoid potential misunderstandings, we have added clarifications to the manuscript to explicitly state that this study is based on an idealized setup and does not intend to mimic operational implementations. Specifically, the following sentences have been added:

- **NODA Experiments Section (Page 20, Lines 599-608):**
  "*For the IOP13, NODA experiment is a direct downscaling from EPS-ECMWF boundary and initial conditions valid at 1800 UTC 14 October to 00 UTC 16 October 2012 (Fig. 6c). The comparison among NODA, CNTRL and SYN will provide us with valuable information on the impact of assimilating different sources of observations.*

  *For Qendresa, NODA experiment is simply a direct downscaling of 36 hours from EPS-ECMWF at 00 UTC 7 November to 12 UTC 8 November 2014 (Fig. 6f). Here again, it is important to note that the choice of starting NODA at 00 UTC 7 November instead of starting at 12 UTC 6 November was made intentionally to extract general conclusions applicable to an operational framework.*"

In the revised version of the manuscript, we have avoided referring to the "operational" framework. We hope this clarification addresses the reviewer's concern and provides a clearer understanding of the study's scope.

**Major Comments:**

*L 79-84: The literature review is lacking, and there are several studies on small-scale convective NWP that impact coastal regions of the Mediterranean, also with data assimilation approaches. It is recommended to expand the literature review on this topic and reconsider the misleading information in these lines.*

We thank the reviewer for pointing out the need to expand the literature review and for highlighting the relevance of studies on small-scale convective NWP and data assimilation approaches in coastal Mediterranean regions. While we agree that the existing literature on Data Assimilation for NWP is vast, we found relatively few studies that specifically focus on **applying Data Assimilation to extreme weather events affecting coastal regions of the Mediterranean**, particularly using EnKF and 3DVar methodologies. Producing accurate forecasts of extreme weather events initiated over maritime regions and affecting populated Mediterranean coastal regions remains a major challenge for the forecasting community. This challenge arises mainly due to the scarcity of observational data, the complex topography surrounding the Mediterranean basin, and the intricate interactions between sea and land processes. Most DA studies available in the literature focus on relatively flat land regions, with very few addressing the Mediterranean basin. This gap underscore the novelty and relevance of the present study.

To address the reviewer's concern, we performed a more extensive search for relevant studies and have identified a few additional papers that align with the scope of our work. We have expanded the literature review to include the following references:

1.  Carrió, D. S., & Homar, V. (2016). *Potential of sequential EnKF for the short-range prediction of a maritime severe weather event*. *Atmospheric Research*, *178*, 426-444.

2.  Amengual, A., Carrió, D. S., Ravazzani, G., & Homar, V. (2017). *A comparison of ensemble strategies for flash flood forecasting: The 12 october 2007 case study in Valencia, Spain*. *Journal of Hydrometeorology*, *18*(4), 1143-1166.

3.  Carrió, D. S., Homar, V., & Wheatley, D. M. (2019). *Potential of an EnKF storm-scale data assimilation system over sparse observation regions with complex orography*. *Atmospheric Research*, *216*, 186-206.

4.  Amengual, A., Hermoso, A., Carrió, D. S., & Homar, V. (2021). *The sequence of heavy precipitation and flash flooding of 12 and 13 September 2019 in Eastern Spain. Part II: A hydrometeorological predictability analysis based on convection-permitting ensemble strategies*. *Journal of Hydrometeorology*, *22*(8), 2153-2177.

5.  Mazzarella, V., Ferretti, R., Picciotti, E., & Marzano, F. S. (2021). *Investigating 3D and 4D variational rapid-update-cycling assimilation of weather radar reflectivity for a heavy rain event in central Italy*. *Natural Hazards and Earth System Sciences*, *21*(9), 2849-2865.

6.  Torcasio, R. C., Federico, S., Comellas Prat, A., Panegrossi, G., D'Adderio, L. P., & Dietrich, S. (2021). *Impact of lightning data assimilation on the short-term precipitation forecast over the Central Mediterranean Sea*. *Remote Sensing*, *13*(4), 682.

7.  Capecchi V, Antonini A, Benedetti R, Fibbi L, Melani S, Rovai L, Ricchi A, Cerrai D. Assimilating X- and S-Band Radar Data for a Heavy Precipitation Event in Italy. Water. 2021; 13(13):1727. https://doi.org/10.3390/w13131727.

8.  Lagasio, M., Silvestro, F., Campo, L., & Parodi, A. (2019). Predictive Capability of a High-Resolution Hydrometeorological Forecasting Framework Coupling WRF Cycling 3DVAR and Continuum. Journal of Hydrometeorology, 20(7), 1307-1337. https://doi.org/10.1175/JHM-D-18-0219.1

If the reviewer is aware of additional studies that meet these criteria, we would greatly appreciate their suggestions and would be happy to include them in our review.

Furthermore, we have carefully reviewed the lines identified by the reviewer for any potentially misleading information and have revised the text accordingly to ensure accuracy and clarity. Specifically, the literature review section has been updated to reflect these new references and to clarify the novelty and focus of our study.

- **Introduction Section (Page 2, Lines 79-84):**
  "*However, less attention has been paid to convective-scale NWP problems, especially those associated with small scale convective phenomena initiated over regions with sparse observational data coverage, such as the extreme weather events affecting coastal regions in the Mediterranean basin (Carrió et al., 2016; Amengual et al., 2017; Carrió et al., 2019; Amengual et al., 2021; Mazzarella et al., 2021; Torcasio et al., 2021).*"

These revisions aim to address the reviewer's concern and ensure that our literature review appropriately reflects the current state of research in this field. We thank the reviewer for this constructive suggestion and believe that these updates will improve the manuscript significantly.

*L149-150 and 155-157: In this case as well, there is a gap in the literature review. Several studies on the Mediterranean region and Italy configure simulations, also with data assimilation and for*

*operational purposes. It is recommended to update the literature review and position the aim of the study and its innovation in relation to the already existing works.*

We thank the reviewer for this observation and appreciate the suggestion to strengthen the literature review.

For **Lines 149-150**, we have expanded the literature review by incorporating additional references related to data assimilation studies in the Mediterranean region, particularly those focused on high-resolution numerical simulations with DA techniques. These additions provide a broader context and better position our study in relation to previous research.

The revised text for **Lines 149-150** now reads as follows (new references are underlined):

"*Recent convective-scale DA studies have primarily focused on the mature stage of weather events (e.g., Tong et al., 2005; Fujita et al., 2007; Dowell et al., 2021; Jones et al., 2013; Wheatley et al., 2015; Jones et al., 2016; Gao et al., 2016; Ballard et al., 2016; Gustafsson et al., 2018; Carrió et al., 2019; Mazzarella et al., 2020; Yussouf et al., 2020; Federico et al., 2021; Junjun et al., 2021; Janjić et al., 2022; Wang et al., 2022).*"

Regarding **Lines 155-157**, we would like to emphasize that the number of studies specifically addressing data assimilation in pre-convective environments are relatively scarce. Most DA studies focus on assimilating observations once the weather phenomena has already initiated, rather than incorporating observations several hours before the event begins (i.e., in the pre-convective phase). Nonetheless, we have included some relevant studies we were able to identify in the revised manuscript. To the best of our knowledge, very few studies focus explicitly on DA strategies in pre-convective conditions, particularly in the Mediterranean region. This distinction underscores the novelty of the present study and its contribution to the existing body of research. If the reviewer is aware of additional references on this topic, we would greatly appreciate their suggestions, and we would be happy to incorporate them into our discussion.

The revised text for **Lines 155-157** now reads as follows:

"*Despite its potential benefits, very few studies have explored the role of DA in the pre-convective stage (e.g., Carrió et al., 2019; Carrió et al., 2022; Corrales et al., 2023), where assimilating observations before convection initiates could significantly improve forecast lead time, providing advanced warnings and allowing decision-makers to act proactively.*"

We have also updated the manuscript to clearly delineate the novelty of our study in comparison to the existing works. Specifically, the literature review section now better contextualizes the aim of our study and its contributions in relation to prior research.

We hope this revision addresses the reviewer's concerns and provides a more comprehensive positioning of our study within the existing body of work.

**New References:**

*Corrales, P. B., Galligani, V., Ruiz, J., Sapucci, L., Dillon, M. E., Skabar, Y. G., ... & Nesbitt, S. W. (2023). Hourly assimilation of different sources of observations including satellite radiances in a mesoscale convective system case during RELAMPAGO campaign. Atmospheric Research, 281, 106456.*

*Tong, M., & Xue, M. (2005). Ensemble Kalman filter assimilation of Doppler radar data with a compressible nonhydrostatic model: OSS experiments. Monthly Weather Review, 133(7), 1789-1807.*

Fujita, T., Stensrud, D. J., & Dowell, D. C. (2007). Surface data assimilation using an ensemble Kalman filter approach with initial condition and model physics uncertainties. Monthly weather review, 135(5), 1846-1868.

Carrió, D. S., Jansà, A., Homar, V., Romero, R., Rigo, T., Ramis, C., ... & Maimó, A. (2022). Exploring the benefits of a Hi-EnKF system to forecast an extreme weather event. The 9th October 2018 catastrophic flash flood in Mallorca. Atmospheric Research, 265, 105917.

Mazzarella, V., Maiello, I., Ferretti, R., Capozzi, V., Picciotti, E., Alberoni, P. P., ... & Budillon, G. (2020). Reflectivity and velocity radar data assimilation for two flash flood events in central Italy: A comparison between 3D and 4D variational methods. Quarterly Journal of the Royal Meteorological Society, 146(726), 348-366.

Jones, T. A., Otkin, J. A., Stensrud, D. J., & Knopfmeier, K. (2013). Assimilation of satellite infrared radiances and Doppler radar observations during a cool season observing system simulation experiment. Monthly weather review, 141(10), 3273-3299.

Federico, S., Torcasio, R. C., Puca, S., Vulpiani, G., Comellas Prat, A., Dietrich, S., & Avolio, E. (2021). Impact of radar reflectivity and lightning data assimilation on the rainfall forecast and predictability of a summer convective thunderstorm in Southern Italy. Atmosphere, 12(8), 958.

Gao, J., Fu, C., Stensrud, D. J., & Kain, J. S. (2016). OSSEs for an ensemble 3DVAR data assimilation system with radar observations of convective storms. Journal of the Atmospheric Sciences, 73(6), 2403-2426.

Dowell, D. C., Wicker, L. J., & Snyder, C. (2011). Ensemble Kalman filter assimilation of radar observations of the 8 May 2003 Oklahoma City supercell: Influences of reflectivity observations on storm-scale analyses. Monthly Weather Review, 139(1), 272-294.

Carrió, D. S., Homar, V., & Wheatley, D. M. (2019). Potential of an EnKF storm-scale data assimilation system over sparse observation regions with complex orography. Atmospheric Research, 216, 186-206.

Janjić, Tijana, Yuefei Zeng, and Yvonne Ruckstuhl. "Weakly Constrained LETKF for Convective-Scale Data Assimilation." (2022).

Gustafsson, N., Janjić, T., Schraff, C., Leuenberger, D., Weissmann, M., Reich, H., ... & Fujita, T. (2018). Survey of data assimilation methods for convective-scale numerical weather prediction at operational centres. Quarterly Journal of the Royal Meteorological Society, 144(713), 1218-1256.

Ballard, Susan P., et al. "Performance of 4D-Var NWP-based nowcasting of precipitation at the Met Office for summer 2012." Quarterly Journal of the Royal Meteorological Society 142.694 (2016): 472-487.

Hu, Junjun, et al. "Assessment of Storm-Scale Real Time Assimilation of GOES-16 GLM Lightning-Derived Water Vapor Mass on Short Term Precipitation Forecasts During the 2020 Spring Forecast Experiment." Journal of Geophysical Research: Atmospheres 126.21 (2021): e2021JD034603.

Wang, Y., Yussouf, N., Kerr, C. A., Stratman, D. R., & Matilla, B. C. (2022). An experimental 1-km Warn-on-Forecast System for hazardous weather events. Monthly Weather Review, 150(11), 3081-3102.

L161-172: Point a) The comparison with the NODA run is missing in order to address this point. Please refer to General Comment 2 concerning the NODA run that should be used to effectively assess the impact of data assimilation. Point b) It is true that assimilation starts in the pre-convective

*phase of the event, but it extends until the beginning of the validated time window. How does this approach differ from other similar methods that have already been published and are commonly used in operational contexts? Point c) To assess the impact of observations with high spatial and temporal resolution, why was a broader domain coverage not used, provided by other denser and more homogeneous networks available in Italy? Point d) Comparisons with the NODA run are missing.*

Point a): We appreciate the reviewer's suggestion and agree that including the NODA run enhances the robustness of our assessment. In response to this comment, we have incorporated the results from NODA runs into our analysis and included new discussions in the manuscript to more effectively evaluate the impact of data assimilation. Please, also refer to our response to **General Comment 2**, where we provided additional details on this aspect.

Point b): Typically, Data Assimilation is applied once the weather event has already initiated and developed. For instance, studies focusing on improving tornadoes forecasts over the U.S. Great Plains often assimilate observations only after the tornado has developed, rather than in the hours leading up to its development. From a Warning System perspective, this approach has limited advantages, as the event is already underway, leaving less time for authorities to issue effective warnings and mitigate potential impacts.

This is precisely why assimilating observations before the event begins – during the pre-convective phase – is crucial. Forecasting the onset of an extreme weather event is significantly more challenging than predicting its subsequent evolution. By implementing this approach, we aim to improve the lead time for early warnings. We have previously applied this methodology in several studies (e.g., Carrió et al., 2016, 2018, 2019), demonstrating its potential benefits in enhancing forecast accuracy for high-impact weather events.

Point c): The observations used in this study to validate our DA simulations were obtained from the publicly available observations provided by MADIS from the *National Oceanic and Atmospheric Administration* (NOAA). While denser observational networks exist in Italy, these datasets are generally not publicly accessible to all users. Our selection of observational data therefore was based on the public available sources to ensure accessibility and reproducibility of the present study.

Point d): As noted in Point a), we acknowledge the reviewer's concern regarding the missing NODA runs. To address this, we have now included the results from NODA runs in the revised version of the manuscript, along with a corresponding discussion to better assess the impact of data assimilation.

*Section 3: The study presents a potentially interesting comparison, but it is based on only two case studies and two sets of observations that differ significantly. The in-situ observation sets also appear to vary considerably between the two cases, which makes it challenging to draw a general conclusion. While it is true that observations over the sea are generally limited, a more homogeneous correction across the entire domain could still lead to improvements in the initial and boundary conditions, and, consequently, in the simulation. As it stands, the two cases seem somewhat disconnected, making it difficult to draw a broader conclusion.*

We thank the reviewer for this comment and appreciate the opportunity to clarify this relevant point.

This study is not intended to derive general conclusions applicable to all scenarios, but rather to provide a focused comparison of two widely used data assimilation methods – EnKF and 3DVar – in the context of extreme weather events that initiated over the sea, where observational data are scarce. We want to assess the potential of these methods in improving predictability under such challenging conditions, rather than to establish statistically robust conclusions across multiple cases. Several studies published in top-ranked journals have adopted a similar methodology, using one or two real-case weather events to assess the performance of different existing DA methods. Notables examples include:

1. Tiwari, G. et al., 2022. *Predictive skill comparative assessment of WRF 4DVar and 3DVar data assimilation: An Indian Ocean tropical cyclone case study. Atmospheric Research*
2. Sun, J. et al., 2013. *Radar data assimilation with WRF 4D-Var. Part II: Comparison with 3D-Var for a squall line over the US Great Plains. Monthly Weather Review*
3. Mazzarella, V., Maiello, I., Capozzi, V., Budillon, G., & Ferretti, R. (2017). *Comparison between 3D-Var and 4D-Var data assimilation methods for the simulation of a heavy rainfall case in central Italy. Advances in Science and Research, 14, 271-278.*
4. Mazzarella, V., Maiello, I., Ferretti, R., Capozzi, V., Picciotti, E., Alberoni, P. P., ... & Budillon, G. (2020). *Reflectivity and velocity radar data assimilation for two flash flood events in central Italy: A comparison between 3D and 4D variational methods. Quarterly Journal of the Royal Meteorological Society, 146(726), 348-366.*

We acknowledge that the two case studies involve different sets of *in-situ* observations, reflecting the inherent variability in real-world data availability over maritime regions. However, this diversity is intentional, as it highlights the challenge of data assimilation in such environments and provides valuable insights into how each method performs under different observational constraints. The key motivation of this study was to select two distinct extreme weather events initiated over the sea to analyze how these two methods behave under significantly different conditions. Unlike many DA studies that focus on well-observed land-based events, our approach deliberately examines extreme weather phenomena in regions with scarce *in-situ* observations. By doing so, we aim to contribute to the understanding of DA performance in maritime environments, a topic that remains underrepresented in the existing literature.

To further clarify this, we have revised the manuscript to explicitly state the study's scope and objectives, ensuring that readers do not misinterpret it as an attempt to generalize findings beyond the examined cases. See previous answers for further details.

We hope this explanation addresses the reviewer's concern and clarifies the intent of the study.

*L447: The period used for constructing the background is only 2 weeks, which is half of the minimum duration recommended in the WRFDA user guide (at least a 1-month dataset). This could negatively impact the comparison between 3DVar and EnKF, potentially disadvantaging the 3DVar. Have sensitivity tests been conducted using B matrices constructed from longer periods?*

We thank the reviewer for the opportunity to clarify this relevant aspect. Several studies in the literature use a two-week period for calculating the B matrix (Hung et al. 2023; Fitzpatrick et al., 2007; Mazzarella et al., 2020, 2021), and all show an improvement in reproducing the initial conditions. It's worth noting that the WRFDA user guide states: "For the NMC-method, the model perturbations are differences between forecasts (e.g. T+24 minus T+12 is typical for regional applications, T+48 minus T+24 for global) valid at the same time. Climatological estimates of background error may then be obtained by averaging these forecast differences over a period of time (e.g. one month)" (WRFDA Users Guide, pp. 41). This clearly suggests that the recommended one-month period is not mandatory, and a shorter duration, such as two weeks, can be used. In the context of this study, we adopted the CV7 option with the default "len scale" and "var scale" parameters. While acknowledging that a sensitivity analysis of these parameters could potentially lead to further improvements in model performance, this was not among the primary objectives of this work.

We hope this explanation satisfactorily addresses the reviewer's concern and enhances the clarity and improves the clarity and completeness of our manuscript.

**New References:**

Hung MK, Tien DD, Quan DD, Duc TA, Dung PTP, Hole LR, Nam HG. Assessments of Use of Blended Radar–Numerical Weather Prediction Product in Short-Range Warning of Intense Rainstorms in Localized Systems (SWIRLS) for Quantitative Precipitation Forecast of Tropical Cyclone Landfall on Vietnam's Coast. Atmosphere. 2023; 14(8):1201. https://doi.org/10.3390/atmos14081201

Fitzpatrick, P. J., Li, Y., Hill, C., Karan, H., Lim, E., & Xiao, Q. (2007, June). The impact of radar data assimilation on a squall line in Mississippi. In 22nd Conference on Weather Analysis and Forecasting/18th Conference on Numerical Weather Prediction.

Mazzarella, V.; Maiello, I.; Ferretti, R.; Capozzi, V.; Picciotti, E.; Alberoni, P.; Marzano, F.; Budillon, G. Reflectivity and velocity radar data assimilation for two flash flood events in central Italy: A comparison between 3D and 4D variational methods. Q. J. R. Meteorol. Soc. 2020, 146, 348–366.

Mazzarella, V., Ferretti, R., Picciotti, E., and Marzano, F. S.: Investigating 3D and 4D variational rapid-update-cycling assimilation of weather radar reflectivity for a heavy rain event in central Italy, Nat. Hazards Earth Syst. Sci., 21, 2849–2865, https://doi.org/10.5194/nhess-21-2849-2021, 2021.

The revised text now reads as follows:

"*Consistent with several papers (Hung et al. 2023; Fitzpatrick et al., 2007; Mazzarella et al., 2020, 2021) showing positive results, a two-week period was used for the calculation of the B matrix in this study*"

*L503-504: A multi-physics approach for the EnKF, compared to a deterministic run with 3DVar, significantly disadvantages the deterministic simulation, regardless of the effect of data assimilation. A comparison between the ensemble and the deterministic simulation without the use of DA should be conducted to effectively evaluate the impact of the two DA methods.*

We thank the reviewer for raising this point. EnKF and 3DVar are fundamentally different Data Assimilation approaches, primarily in how they estimate the **background error covariance matrices**, which play a crucial role in their performance. Comparisons between EnKF and 3DVar/4DVar are widely conducted in the literature, as they provide valuable insights into the strengths and limitations of each method under different scenarios.

The EnKF methodology inherently relies on an **ensemble-based approach** to estimate **flow-dependent background error covariances**, ensuring that error structures evolve dynamically with the forecast. In contrast, 3DVar employs **static, climatological background error covariances**, which do not evolve with time but can be tuned based on long-term statistics. The use of a multi-physics ensemble in EnKF is a standard practice, as it ensures sufficient ensemble spread and better representation of background error statistics, which are essential for the proper functioning of the EnKF system. Running EnKF with a deterministic configuration similar to 3DVar would significantly degrade its performance, as it would lack the necessary variability to correctly compute error covariances.

Here, it is important to highlight that, due to differences in how background error covariances are computed, 3DVar can sometimes produce more realistic background error covariance estimates over the sea than an ensemble-based system like EnKF. This is because EnKF relies on short-range ensemble forecasts to approximate these covariances, whereas 3DVar can incorporate climatological information, which may be particularly advantageous in data-sparse maritime regions.

It is not among the aims of this paper the comparison between the ensemble simulation and the deterministic one. There are many papers addressing this problem.

We hope this clarification helps address the reviewer's concern and provides further insight into the methodological choices in our study.

*L530-531 and 547-548: why the reference experiment is not included in the validation? It is impossible to assess the impact of the assimilation without a reference simulation. Where is the NODA experiment in the work results analysis?*

We thank the reviewer for this observation. We fully agree that including a reference experiment is essential for assessing the impact of data assimilation. In response to this comment, we have now incorporated NODA results into our analysis and added a corresponding discussion in the manuscript.

These additional results allow for a clearer evaluation of the improvements introduced by data assimilation, ensuring a more comprehensive assessment of the relative performance of the EnKF and 3DVar methods.

We hope this revision adequately addresses the reviewer's concern and enhances the clarity and completeness of our results analysis.

*L553-555: The comparison is certainly of interest. However, it is based on only two case studies, where two different sets of observations were used in the CNTRL simulation for two extreme events with completely different characteristics. This makes it challenging, if not impossible, to draw general conclusions. Typically, the run that uses high spatial and temporal resolution data, such as radar and satellite products, shows a greater improvement compared to one that only uses in-situ observations, which is almost well-established in the literature. Expanding the analysis to include more cases or a more consistent dataset could strengthen the conclusions drawn.*

We thank the reviewer for this comment and appreciate the opportunity to further clarify the scope of our study. As previously mentioned in our responses, this study is not intended to derive general conclusions but rather to provide a focused comparison of two widely used data assimilation methods – EnKF and 3DVar – when applied to extreme weather events initiated over the sea, where observational data are scarce.

We acknowledge that the two case studies involve different sets of in-situ observations, reflecting the inherent variability in real-world data availability over maritime regions. However, this diversity is intentional, as it highlights the challenges of data assimilation in such environments and provides insights into how each method performs under different observational constraints. Additionally, studies that compare different DA methods based on one or two real-case experiments are common in the literature and have provided valuable contributions to understanding DA performance.

While we agree that expanding the analysis to include more cases could further generalize the findings, the goal of this study is not statistical validation but rather a methodological comparison under contrasting extreme weather conditions. The selection of these two cases was made in purpose, as they represent significantly different meteorological scenarios, allowing us to assess the strengths and limitations of each DA method across different conditions.

To avoid any misinterpretation, we have revised the manuscript to further clarify the study's objectives and limitations, ensuring that readers do not mistakenly assume that broad generalizations are being drawn from the results.

We hope this explanation satisfactorily addresses the reviewer's concern.

*L612-613: As mentioned in general comment 4, if the aim is to assess the applicability to an operational framework, the simulation time required for these configurations should be included in the analysis. This would help determine whether the validation time window chosen for the results analysis is fully available as a forecast or if part of it will be in hindcast considering that in real time*

*the assimilation should wait for the observations availability. For operational applications, it is crucial that the forecast be provided in advance of the event's time window, ideally by at least a few hours. While this may be negligible in the case of a long-lasting event, as in the second use case presented in the work, it is fundamental for the first use case.*

We thank the reviewer for this comment and appreciate the opportunity to clarify this point once again. As previously stated in our response to General Comment 4, **this study is not designed as an operational application** but rather as a methodological evaluation of two widely used data assimilation techniques under optimal conditions. Therefore, assessing the computational time required to run these experiments is not within the scope of this study.

Our primary objective is to analyze the potential of these DA methods in improving the predictability for extreme weather events originating over the sea, where the observational data are scarce. To achieve this, we assume a perfect scenario where computational constraints are not a limiting factor. While we acknowledge that operational applications must account for real-time processing constraints and observational availability, these aspects fall beyond the intended scope of this work.

To prevent further misunderstanding, we have added a clarification in the manuscript explicitly stating that this study does not aim to evaluate operational feasibility but rather to compare the performance of these DA methods under optimal conditions.

We hope this explanation satisfactorily addresses the reviewer's concern.

*Figure A1: For the evaluation of hourly rainfall, only the 1-hour forecast between 00:00 and 01:00 on the 15th was considered, which corresponds to the moment when the data assimilation has just finished. It would be more appropriate to evaluate all hourly accumulations between 00:00 and 06:00 for consistency with the 6-hourly evaluation and to provide a more comprehensive view of the period during which data assimilation has an impact.*

We thank the reviewer for this valuable suggestion. In response to this comment, we have updated the figure to include the evaluation of all hourly rainfall accumulations between 00:00 and 06:00, ensuring consistency with the 6-hourly evaluation and providing a more comprehensive view of the period during which data assimilation has an impact.

The updated figure has been included in the **Section "New Figures"** of this document.

*L787-789: The members with poorer performance use the Kain-Fritsch cumulus parametrization, which is the same as the deterministic simulation employed for the 3DVar. Was an attempt made with 3DVar, using the Grell-Freitas cumulus parametrization in combination with the YSU PBL scheme, which appears to be the best combination in the ensemble, to assess the impact of this difference?*

We thank the reviewer for this comment. However, we would like to clarify that this study is not designed as a sensitivity analysis of physical parameterizations, but rather as a comparison between two widely used data assimilation methods – 3DVar and EnKF – in their established configurations.

The choice of physical parameterizations in each system follows commonly used configurations for operational and research application. The 3DVar experiment employs a deterministic configuration, while the EnKF system relies on an ensemble-based approach with multi-physics to ensure sufficient spread and proper representation of flow-dependent background error covariances. Since the objective of this study is to evaluate the performance of EnKF and 3DVar under their respective standard implementations, we did not modify the physical parameterizations in the 3DVar experiment. A separate sensitivity study on parameterization schemes would be required to specifically address this question, but it is beyond the scope of the present work.

We hope this clarification addresses the reviewer's concern.

*Summary and Conclusions section: The conclusions of this work should be revisited in light of the previous comments and the work that still needs to be implemented.*

We thank the reviewer for this comment and appreciate their constructive feedback through the review process. In response to the reviewer's suggestions, we have revised the Summary and Conclusions section to reflect the updates made to the manuscript, including the incorporation of NODA runs, the extended hourly rainfall evaluation, and the clarification of the study's scope and objectives.

These revisions ensure that the conclusions accurately summarize the findings while maintaining consistency with the additional analyses and discussions implemented throughout the manuscript.

We hope the revised conclusions now better align with the reviewer's expectations and the updated content of the study.

**Minor comments:**

*L137-138: Given the approach proposed for an operational purpose, it is recommended to include a more detailed comparison of computational efforts and timing in some section of the paper. This could be important for further evaluating the choice of using an EnKF approach compared to the deterministic 3DVar.*

We thank the reviewer for this suggestion. However, as previously clarified in General Comment 4 and L612-613, **this study is not designed for operational implementation** but rather as a methodological comparison of two widely used data assimilation approaches – EnKF and 3DVar – under optimal conditions. Therefore, a detailed comparison of computational costs and timing was not within the scope of this work.

We acknowledge that computational efficiency is a crucial factor for operational applications. However, evaluating the feasibility of these approaches in an operational setting would require additional considerations, including hardware specifications, model resolution, data availability latencies, and system optimization, which are beyond the focus of this study.

To avoid potential misunderstandings, we have revised the manuscript to explicitly **clarify that this study does not aim to assess operational feasibility,** but rather to explore the potential of these DA methods in improving predictability in maritime extreme weather events.

We hope this clarification adequately addresses the reviewer's concern.

*L174-175: If "this study is not aimed to draw any statistically significant conclusion", how can points a, b, c, and d be addressed?*

We thank the reviewer for this observation and appreciate the opportunity to clarify this point. The statement "*this study is not aimed to draw any statistically significant conclusion*" refers to the fact that our analysis is **not based on a large sample of cases** and does not attempt to establish generalized statistical relationships. The statistical framework is only used to objectively compare the two methods. Instead, the study focuses on a methodological comparison between 3DVar and EnKF in two distinct extreme weather events that originated over the sea, where observational data are scarce.

However, despite the limited number of case studies, the points (a, b, c and d) are still addressed by qualitative and process-based evaluations rather than statistical inference. The methodology used allows us to assess the behavior of each DA approach under different conditions and to provide insights into their potential advantages and limitations in such challenging environments.

To avoid potential misinterpretation, we have revised this statement in the manuscript to clarify that while the study does not aim to establish statistically significant findings, it does provide a comparative assessment of DA methodologies based on two well-defined case studies. For additional clarification, the reviewer may refer to our response to the first "General Comment", where we have reformulated the study objectives (points a, b, c and d) to better reflect the scope of this research.

We hope this explanation resolves the reviewer's concern.

*L259-266: Italy has a dense national network of both radar and in-situ stations, offering comprehensive coverage across the territory. Could you please clarify why the decision was made not to utilize all available observations, which would provide more homogeneous and dense coverage over the entire domain of interest, potentially enhancing the data assimilation process? Especially in the Qendresa use case, was there no radar available with at least some coverage over the sea?*

We thank the reviewer for this comment and appreciate the opportunity to clarify our data selection process.

The observational data used in this study were obtained from publicly available datasets, specifically from MADIS (NOAA), ensuring accessibility and reproducibility of our results. While we acknowledge that Italy has a dense observational network, **not all datasets are openly accessible** for research purposes. Many high-resolution national networks have **restrictive access**, limiting their availability to external users.

Regarding the Qendresa case, reflectivity observations from radar were not available for the dates and locations relevant to the event. Additionally, the southernmost radar in Italy does not provide sufficient coverage over the initiation region over the sea, where Qendresa developed. Furthermore, even if some radar observations were available, they are **not publicly accessible and lack quality control**, which could introduce uncertainties into the data assimilation process. In contrast, for the IOP13 case, we used radar data that were quality-controlled and archived on the **HyMeX** website, ensuring reliability.

This study focuses on assessing DA performance in maritime-origin extreme weather events, where observational constraints are a major challenge. The dataset used reflects **realistic limitations in data availability** when forecasting such events. However, we recognize that incorporating a denser, more homogeneous observational network could further improve DA performance. Future studies could explore the impact of assimilating additional observations, if accessible, to evaluate their contribution to forecast improvements.

The revised text for **Section IOP13 Observations** now reads as follows:

[revised manuscript text omitted]

We hope this explanation clarifies our approach and the reasoning behind our data selection.

*L437-439: The observation operator used considers only the warm rain process and not all the ice microphysics species. In the case of extreme convective events, ice species are typically an important component of the cloud, as evidenced by the usual presence of lightning. Has the impact of using a warm rain approach on radar data assimilation been evaluated, especially in the 3DVar run, which has a single physical configuration due to its deterministic nature?*

We thank the reviewer for this insightful comment. We acknowledge that in extreme convective events, ice species play a crucial role, as evidenced by the frequent presence of lightning. However, the primary focus of this study is to compare the performance of 3DVar and EnKF data assimilation techniques using the default setup, rather than to conduct a detailed sensitivity analysis of different microphysical assumptions within the observation operator. Assessing the specific impact of using a warm rain approach versus a more complete microphysics scheme is an important topic but falls beyond the scope of the present study. Additionally, note that the observation operator used in this study is the same used in previous papers (i.e., Carrió et al., 2019; Mazzarella et al., 2020), where it

has been successfully applied in similar contexts. Therefore, this choice is consistent with established practices in convective-scale data assimilation studies.

We appreciate this suggestion and acknowledge that future work could explore the impact of assimilating radar reflectivity with an observation operator that includes ice-phase processes, particularly for extreme convective events. We have added a brief discussion in the revised manuscript to acknowledge this limitation and highlight it as a potential direction for future research.

We hope this clarification addresses the reviewer's concern.

*L467-478: The domains chosen in this work are quite small and one-way nested, which may limit the duration of the data assimilation effect. Given the operational purpose emphasized in the study, why was a single, national-scale domain not used? This approach could have also facilitated a more homogeneous comparison between the two case studies and potentially allowed for a longer data assimilation impact. Can you justify also the use of a one-way nested approach instead of a two-way one?*

We thank the reviewer for this comment and appreciate the opportunity to clarify our choices regarding the domain configuration and nesting approach.

First, it is important to note that using nesting domains is the most common practice in convective-scale DA studies (e.g., Snyder and Zhang, 2003; Dowell et al., 2004; Tong and Xue, 2005; Fujita et al., 2007; Wheatley et al., 2012; Yussouf et al., 2015, Carrió et al., 2019). The domains used in this study were carefully designed to balance high-resolution representation of convection with computational feasibility. The small, high-resolution inner domains were chosen to focus on the areas directly affected by the extreme weather events, ensuring that data assimilation had a meaningful impact on the convective-scale processes. Using a single national-scale domain, while beneficial for consistency across cases, would have required a coarser resolution to maintain computational efficiency, potentially reducing the benefits of data assimilation at convective scales. Additionally, using a nested-domain approach allows for a more effective treatment of the **multiscale data assimilation problem**. The assimilation of *in-situ* observations primarily enhances the estimation of large-scale atmospheric patterns, while higher-resolution observations, such as radar data, contribute to improving small-scale weather features. This distinction is crucial in high-resolution numerical weather prediction and justifies the use of nested domains in this study. Furthermore, the domain sizes selected are comparable to those used in previous research, reinforcing the validity of this approach. As previously clarified, this study is not intended for operational implementation, but rather to evaluate the potential of EnKF and 3DVar in extreme weather scenarios, making a national-scale setup less relevant for our research objectives.

Regarding the one-way nested approach, this choice was made to ensure numerical stability and prevent feedback loops between the large-scale and fine-scale domains. A two-way nesting approach allows information from the inner domain to influence the outer domain, which may introduce inconsistencies when performing high-resolution data assimilation experiments. In our setup, one-way nesting ensures that the data assimilation process does not artificially influence the larger-scale meteorological fields, maintaining a cleaner evaluation of the data assimilation impact within the targeted high-resolution domain. Additionally, based on previous experience running nested high-resolution simulations, we have observed that two-way nesting is significantly more computationally demanding. Given the computational cost-benefit tradeoff and the focus of this study on evaluating convective-scale data assimilation, a one-way nesting approach was deemed the most appropriate choice for maintaining numerical consistency and computational efficiency.

We recognize that alternative domain configuration, including a single large-scale domain or two-way nesting, could provide different insights. However, the current configuration was selected based on best practices for convective-scale DA studies and the specific objectives of this work.

We hope this explanation adequately addresses the reviewer's concern.

*L564-565: Several studies have shown that the assimilation of high spatial and temporal resolution observations helps to reduce model spin-up. Could you clarify the reasoning behind the decision to use the first 6-hours of simulation without DA, rather than utilizing the DA to reduce spin-up and potentially shorten the simulation time?*

We thank the reviewer for this insightful comment and appreciate the opportunity to clarify our methodology.

The decision to exclude data assimilation during the first 6 hours of simulation was made to allow the model to achieve dynamical balance before initiating the assimilation process. While it is true that assimilating high spatial and temporal resolution observations can help reduce model spin-up in some cases, it is also important to ensure that the model fields are sufficiently adjusted to the mesoscale environment before introducing observational corrections.

One of the key challenges in convective-scale data assimilation is that assimilating observations too early in the model integration can introduce inconsistencies if the initial fields are still undergoing significant adjustments. By allowing an initial period without assimilation, we ensure that the model has time to develop a physically consistent state, reducing the risk of introducing imbalances that could negatively impact the subsequent assimilation cycles.

Furthermore, the 6-hour spin-up period aligns with common practices in previous studies (e.g., Schwartz et al., 2014; Carrió et al., 2019; Mazzarella et al., 2020), where a short free-run period before DA is used to improve the stability and realism of subsequent assimilation cyclones. This approach was found to provide a better balance between model initialization and the effective impact of DA.

We acknowledge that alternative configurations, including an earlier assimilation start time, could be explored in future studies. However, given the focus of this work on evaluating DA performance, we believe that the chosen setup is appropriate for the objectives of this study.

We hope this explanation adequately addresses the reviewer's concern.

*L696: The threshold used for the FSS is 1mm/h. Such a low threshold is not very appropriate to assess extreme precipitation events. It would be more suitable to use progressively higher thresholds to evaluate whether the simulations accurately predict the more intense part of the event.*

We thank the reviewer for this valuable suggestion. We agree that using higher precipitation thresholds for the FSS would provide a more meaningful assessment of the model's ability to capture the more intense portions of the extreme precipitation events.

In response to this comment, we have performed additional analyses using progressively higher thresholds and have updated the relevant figures accordingly. The revised figures are included below in the Section "New Figures" of the present document, and the corresponding discussion has been added to the manuscript to reflect these updates.

We appreciate the reviewer's input, as this improvement enhances the robustness of our assessment of extreme precipitation predictability.

*L716-716: The short impact of reflectivity assimilation could be due to factors such as the observation coverage of the domain, the choice of domain size, and the nesting approach. A different setup might allow for a longer impact of the assimilation.*

We thank the reviewer for this comment. The short-lived impact of reflectivity assimilation is a well-known limitation in DA research (e.g., Gustafsson et al., 2018; Carrió et al., 2019) and it can be attributed to two key factors:

1. **Intrinsic limitations of current DA methodologies**: Most DA schemes, including 3DVar and EnKF, are formulated under the assumption of Gaussian error distributions and linear observation operators. However, radar reflectivity observations are inherently non-Gaussian and highly nonlinear, making their assimilation more challenging. This mismatch can lead to a rapid loss of assimilation impact, as the corrections introduced by reflectivity DA tend to be quickly adjusted by the model dynamics.

2. **Influence of lateral boundary conditions**: Radar observations typically provide coverage over limited portions of the high-resolution domain, as the inner domain size must remain computationally feasible. Consequently, as the model runs freely after the last DA cycle, the newly introduced information from reflectivity assimilation can be gradually replaced or dissipated due to inflowing large-scale information from the lateral boundaries.

While factors such as observation coverage, domain size, and nesting approach can influence the persistence of reflectivity assimilation effects, the fundamental challenge remains the inherent limitations of current DA methods in handling highly nonlinear and non-Gaussian observation operators. However, we emphasize that the objective of this study is not to find the best DA setup for these case studies, but rather to compare two different DA methods using default configurations. A detailed sensitivity analysis on model domain size or assimilation settings falls beyond the scope of this work.

We acknowledge that improving the impact duration of reflectivity DA is an active area of research, with potential solutions including advanced hybrid DA techniques, adaptive covariance tuning, and machine-learning-based observation operators. We have added a brief discussion in the revised manuscript to clarify this aspect and emphasize that this limitation is intrinsic to existing DA frameworks rather than solely a consequence of domain or nesting choices.

*L742-743: The EnKF can benefit from the multi-physics approach when assimilating observations that directly impact the model's microphysical species. For the deterministic 3DVar, was the physical configuration selected the one that performed best for the specific case study?*

We thank the reviewer for this comment. As previously mentioned in our response to an earlier comment, the physical configuration used in the 3DVar experiment was not chosen based on its optimal performance for the specific case study, but rather follows the default configuration used by CETEMPS for 3DVar in this setup. Modifying this setup to optimize performance for a particular case would introduce an additional variable, making it more difficult to isolate the impact of the DA methodology itself. The goal of this study is to compare the performance of two established DA methods rather than to conduct a sensitivity analysis of physical parameterizations within 3DVar.

To avoid redundancy, we refer the reviewer to our previous response addressing this aspect. We hope this clarification resolves the concern.

*L748-750: It would be valuable to include an evaluation and comparison with the ensemble run that uses the same physical setup as the 3DVar.*

We thank the reviewer for this comment. However, as previously explained, configuring the EnKF ensemble to use the same physical setup as the deterministic 3DVar run would significantly degrade the performance of the EnKF system. This is because ensemble variance is crucial for properly estimating flow-dependent background error covariances, which are essential for the effectiveness of EnKF.

Reducing the physical diversity within the ensemble would lead to a dramatic reduction in ensemble spread, increasing the risk of filter divergence, where the system becomes overconfident in its background state and fails to properly adjust to new observations. To prevent this, it is standard practice in ensemble-based data assimilation to incorporate multi-physics or other perturbation strategies to maintain adequate ensemble variability.

The key objective of this study is to compare two established DA methodologies in their respective standard configurations, not to artificially align their setups at the expense of methodological integrity. Additionally, configuring EnKF with a single physical scheme identical to 3DVar would not only compromise the ensemble's performance but also fail to contribute any new scientific insights to the community, as the limitations of such an approach are already well-documented in the literature.

We hope this clarification adequately addresses the reviewer's concern.

*Figure 8: It would be better if RMSE and FSS were applied to the same time window.*

We thank the reviewer for this suggestion. We agree that applying RMSE and FSS to the same time window will improve the consistency of the evaluation. In response to this comment, we have revised Figure 8 to ensure that both metrics are computed over the same time period.

The updated figure has been included in the Section "**New Figures**" below, and the corresponding discussion has been modified accordingly in the main manuscript.

We appreciate the reviewer's input, as this adjustment enhances the clarity and comparability of the results.

*L775-776: It would be preferable if the diagrams covered the same time window as the FSS and RMSE.*

We thank the reviewer for this comment. We agree that Taylor diagrams should use the same time window as FSS and RMSE to ensure consistency across the evaluation metrics. In response, we have updated the Taylor diagrams to align with the same time period. The revised figures have been included in the Section "**New Figures**".

We appreciate the reviewer's suggestion, as this adjustment improves the comparability and coherence of our analysis.

**New Figures:**

**Fig. 6: I have modified the figure according to reviewer's comments.**

[Figure]

[Figure]

**Fig. 8: I have added results from NODA simulation.**

**1h Accumulated precipitation (threshold: 1mm/h):**

[Figure]

**1h Accumulated precipitation (threshold: 3mm/h):**

[Figure]

**3h Accumulated precipitation (threshold: 5 mm/h and 10 mm/h):**

[Figure]

**6h Accumulated precipitation (threshold: 3mm/h):**

[Figure]

[Figure]

**Figure 8**. *Upper panels: Evolution of the FSS during the first 24-h hours of free forecasts for 3-h accumulated precipitation in the Italian sub-regions: R1 (first column), R2 (second column) and R3 (third column). Two thresholds are used: > 5 mm·h⁻¹ (first row) and > 10 mm·h-1 (second row). Lower panels: Evolution of the RMSE associated with each experiment during the first 24 hours of free forecast in the different sub-regions. Simulations assimilating both conventional and radar observations (CNTRL) and simulations assimilating only conventional observations (SYN) associated with the 3DVar and the EnKF are shown here. As a reference, NODA results are also included.*

**Fig. 9: I have added results from NODA (Deterministic) simulation.**

[Figure]

**Figure 9**. *ROC curves and AUC associated with the 3DVar (red and pink colors), EnKF (blue and cyan colors) and NODA (green color) for the 3-hour accumulated precipitation using (a) 1 mm and (b) 10 mm threshold and 6-hour accumulated precipitation using (c) 1 mm and (d) 10 mm threshold, computed over the entire inner domain.*

**Fig. 10: I have added results from NODA simulation.**

[Figure]

**Figure 10**. *Taylor diagram comparing the performance of 3DVar (red), EnKF (blue) and NODA (green) for the 6-hour (left panel) and 24-hour (right panel) accumulated precipitation, valid at 06 UTC 15 October 2012.*

**Fig. 11: I have added results from NODA simulation.**

[Figure]

**Figure 11**. *Temporal surface pressure evolution at the closes grid point to Malta for the (a) SYN and (b) CNTRL experiments associated with the EnKF (blue), 3DVar (red) and NODA (green), compared to the observed surface pressure registered by METARs in Malta's airport (black line).*

**Fig. 12: I have added results from NODA simulation.**

[Figure]

**Figure 12**. *Whisker plots depict the lagged correlation values between the observations and the EnKF (blue boxes), the 3DVar (red stars) and NODA (green stars) for the (a) SYN and (b) CNTRL experiments. The correlation is computed considering that the observed V-shape pressure signature associated with the observations is shifted 4 hours to the left and 4 hours to the right.*

**Fig. 13: I have added results from NODA simulation.**

[Figure]

**Figure 13**. *Probability of cyclone center occurrence computed using Gaussian KDE for (a) NODA, (b) EnKF (SYN), (c) 3DVar (SYN), (d) EnKF (CNTRL) and (e) 3DVar (CNTRL), from 11 UTC 7 November to 12 UTC 8 November 2014. Qendresa's trajectory observed via satellite imagery is depicted in black.*

**Fig. A1**

[Figure]

**Fig. A1**. *1-h accumulated precipitation computed from 00-06 UTC 15 October 2012 associated with Observations (first column), NODA (second column), EnKF (CNTRL) (third column) and 3DVar (CNTRL) (fourth column).*

**Fig. A2**

[Figure]

**Fig. A2**. *6-h accumulated precipitation computed from 00-06 UTC 15 October 2012 associated with (a) Observations, (b) NODA, (c)EnKF (CNTRL), (d) 3DVar (CNTRL).*

---

## Referee Report (RR1)

**Review of "High-Resolution Data Assimilation for Two Maritime Extreme Weather Events: A comparison between 3DVar and EnKF"**

by Diego Saúl Carrió, Vincenzo Mazzarella, and Rossella Ferretti.

The authors investigate potential methods for increasing forecasts and forecast lead times of mesoscale cyclones originating over the Mediterranean sea. Specifically, they compare 3D-Var and EnKF, and evaluate the added value of radar reflectivity and atmospheric motion vectors.

I am pleased with the revisions, which fully addressed the points raised and makes the outcomes of the study clear. Now the manuscript is in good shape and I recommend minor textual revisions, with no need for another review round.

**Minor comments**

It could be noted that some predictability was already present in the noDA experiment since ECMWF performed assimilation at a somewhat larger spatial scale?

L143: Wording: "solve" should probably mean "resolve"?

L148: Rephrase "In this context, which DA method is more suitable?".
For example: "Given limited computational resources, it is unclear which DA method is more accurate."   However, as you know, the answer might depend on how big the resources are.

L427-428: "The assimilation of each observation results in a reduction of the ensemble spread, attributed to using a reduced-moderate ensemble size"
Confusing. Assimilating an observation reduces the analysis variance in variational and Kalman filter assimilation methods. If you want to motivate adaptive inflation, you could say that the small/finite ensemble sizes shrink the ensemble spread more than it should.

L429ff: Which technique was applied: spatially varying or homogeneous? According to the citation it was spatially varying?

Fig 8(i): Data for 3DVar CNTRL is missing.
Lower panels: RMSE unit is mm/h?

Fig 13: Caption: "Probability of cyclone center occurrence" add for example "(within 20 km)"
Check if the values on the colorbars are correct: are all values below 1% or is 0.16 actually 16%?
Tick values should appear once on the colorbar. I guess there is a rounding in place.

L980-981: Ensemble members are deterministic forecasts, right? If so, replace "deterministic numerical weather models" by "the NODA forecast".

L1060: "much less" ... can you quantify that approximately?

L1062-1034: "it does not need either to simulate model trajectories between the assimilation of a set of observations at time t1 and the subsequent set of observations valid at t2"
Confusing. Do you mean that 3D-Var simulates one model trajectory, while the EnKF needed to simulate 36 trajectories?

L900-903: "very different" feels vague. Better to be specific.
"some members could completely fail in the prediction of the weather event": Does it mean that some members did not predict the existence of a cyclone but just unorganized convection?

L903-904: " In this situation, our small-to-moderate ensemble will probably produce a poor flow-dependent background error covariance matrix": If it is somewhat probable that there is no cyclone, then this information should be in the background error covariance, I would say. However, you might mean that large uncertainty/spread leads to substantial nonlinearity, which is detrimental to the analysis accuracy of the EnKF.

L905-907: " for which the ensemble mean will be smoothed out significantly". The ensemble mean should not be expected to be a good forecast of the true state, in case the distribution is non-Gaussian, which it will be for extreme precipitation and the cyclone's pressure field.

L909-911: it is important to note that **although** the ensemble mean of the EnKF_SYN is not correctly reproducing the intensification of Qendresa, some of the ensemble members very well reproduce the observed MSLP both in deepening and timing"
In the light of the above comment, it should not be unexpected that averaging removes extreme values.

L1069: "the 3DVar performs better than the EnKF ensemble mean"
Yes, but then again, this is likely for extreme events because the ensemble mean is averaging over the skewed probability density function. I suggest rephrasing, since obviously in probabilistic metrics, like ROC/AUC, the situation is reversed.

---

## Author Response (AR2)

**Reviewer #1:**

**Minor Comments:**

*The authors have considered all of the remarks by the reviewers. The manuscript has been very much improved, in particular through the following actions:*

*1. Addition of many relevant references*
*2. Better description of aims and their differences with previous studies*
*3. Addition of a table describing experiments and assimilated observations*
*4. Incorporation of no observations data assimilation (NODA) experiments in the evaluation*
*5. Discussion focused on the well-defined aims.*

*I believe the manuscript can be published after the following remark is taken care of:*
*Figure 9: Why are there different number of points on the curves for different experiments? This artificially affects the AUC values.*

We thank the reviewer for the positive feedback and for the constructive comments that have helped us improve the manuscript.

Regarding the comment on Figure 9, we apologize for the confusion. In fact, **all experiments use the same set of thresholds**, resulting in an identical number of ROC points across runs.. The apparent difference arises because, in some experiments, the ROC points cluster very closely near the origin (i.e., at low probability thresholds), causing them to overplot and appear as fewer distinct markers.

To address this and improve clarity, we have **added a note** in the figure caption explaining that all experiments use the same thresholds and that clustering of points reflects tightly grouped ROC values rather than differing point counts.

Now, the figure caption of Figure 9 reads as:

*"**Figure 9**. ROC curves and AUC associated with the 3DVar (red and pink colors), EnKF (blue and cyan colors) and NODA (green color) for the 3-hour accumulated precipitation using (a) 1 mm and (b) 10 mm threshold and 6-hour accumulated precipitation using (c) 1 mm and (d) 10 mm threshold, computed over the entire inner domain. **Note:** all experiments employ the same set of probability thresholds; any apparent differences in the number of plotted points arise from clustering of ROC values at similar thresholds, not from differing data counts."*

We trust these changes resolve the concern.

Again, we want to appreciate the reviewer's feedback, as it has significantly improved the quality of the manuscript.

**Reviewer #2:**

**Major Comments:**

*The literature has now been sufficient extended and includes more works on the topic. However, in lines 152–186, the authors state:*

*"However, at this stage, the system is already well-developed and likely impacting the population, limiting the effectiveness of DA in terms of forecast lead time. In such cases, the potential for early warnings and mitigation actions is significantly reduced, as there is little time left to respond and minimize socio-economic impacts. Despite its potential benefits, very few studies have explored the role of DA in the developing stage (e.g., Carrió et al., 2019; Carrió et al., 2022; Corrales et al., 2023), where assimilating observations before convection initiates could significantly improve forecast lead time, providing advanced warnings and allowing decision makers to act proactively."*

*I do not fully agree with this statement in its current form, as several studies in the literature have already addressed the improvement of forecasts through the assimilation prior to convection initiation, including in coastal Mediterranean areas. Otherwise, the practical value of such forecasts would be minimal or negligible.*

We thank the reviewer for this comment and appreciate the opportunity to clarify this point. Our intent was to motivate this study, highlighting that most of the DA studies at convective-scales are focussed on the mature stage of convective weather events over well-observed land regions. In contrast, DA studies focussed on the development of convective systems over maritime areas, where observations are scarce, remain relatively scarce. One of the aims of this study is to fill this gap by comparing the performance of two widely used DA methods under these data-sparse, maritime conditions. To the best of our knowledge, only a handful of studies have explored analogous setups, combining extreme-case pre-convective assimilation, maritime observation constraints, and high-resolution modeling.

We have adjusted the manuscript to make this distinction clearer and to acknowledge existing work while emphasizing the novelty of our focus on pre‑convective DA in maritime extreme events.

*"However, at this stage, the system is already well-developed and likely impacting the population, limiting the effectiveness of DA in terms of forecast lead time. In such cases, the potential for early warnings and mitigation actions is significantly reduced, as there is little time left to respond and minimize socio-economic impacts. Despite its potential benefits, only a handful of studies have explored the impact of DA using high-resolution numerical models in the developing stage (e.g., Carrió et al., 2019; Carrió et al., 2022; Corrales et al., 2023), and even fewer have done so over data-sparse maritime regions, where early assimilation could be most valuable, providing advanced warnings and allowing decision-makers to act proactively. This study fills that gap by directly comparing two widely used DA techniques – 3DVar and EnKF – in high-resolution, pre-convective assimilation experiments for two extreme weather events initiated over the sea affecting populated coastal regions in the Mediterranean basin."*

*Moreover, in point (b) among the stated objectives, the authors write:*

*"Investigate the potential of using 3DVar and EnKF in the developing phase, that is hours before the mature stage of convective systems are reached, to improve forecast lead time and warning capabilities for extreme weather events."*

*This may be a matter of how the work methodology is currently structured and presented, but as it stands, the manuscript does not clearly highlight a substantial difference from existing studies. I recommend specifying how many hours before the mature phase of the event the final assimilation cycle takes place. This would help demonstrate that assimilation is indeed performed well ahead of convection initiation. Previously, part of the validation was carried out only for the first hour immediately following the last assimilation cycle, which gave the impression that the forecast was not really anticipating the event by much. This may simply be clarified by better defining when the intense phase of the event begins with respect to the last assimilation cycle.*

We thank the reviewer for this suggestion. To clarify the timing of our pre-convective assimilation, we have added the lead time relative to the onset of the mature convective phase. In the revised manuscript, Objective b) now reads as:

*" b) Investigate the potential of using 3DVar and EnKF in the developing phase, specifically 12 hours before the mature stage of convective systems are reached, to improve forecast lead time and warning capabilities for extreme weather events."*

*Based on the simulations setup shown in Figure 6, assimilation continues for 24 hours before the start of the free forecast. Therefore, in the context of "improve forecast lead time and warning capabilities for extreme weather events", what matters is not the initialization time of the forecast itself, but the timing of the final assimilation cycle. In a "warning capabilities" perspective, the forecast would only be available after all observations have been acquired and assimilated, that is, after the last cycle.*

We thank the reviewer for this clarification. To avoid confusion, we have removed "warning capabilities" and focused Objective (b) solely on forecast lead time. The revised objective now reads:

*" b) Investigate the potential of using 3DVar and EnKF in the developing phase, specifically 12 hours before the mature stage of convective systems are reached, to improve forecast lead time."*

*In my view, this remains a crucial point to clarify or change before the publication can be accepted. If the last assimilation cycle is only a couple of hours ahead of the onset of the event, many studies have already investigated similar setups. In that case, I would suggest placing less emphasis on this specific aspect and focusing more directly on the comparison between 3DVar and EnKF, possibly removing this part from the discussion. Alternatively, rather than limiting the analysis to the forecast phase, the authors could consider including an investigation of the pre-convective stage during the assimilation cycles. This would help demonstrate how assimilation improves the model state and how cycling gradually corrects the model, thereby reducing the propagation of errors into the subsequent forecast.*

We thank the reviewer for this insightful recommendation. In light of concerns about manuscript length and scope, we have removed the emphasis on the pre-convective assimilation window from our

discussion and refocused the objectives on the core comparison of 3DVar and EnKF forecast performance. The revised objective now reads as follows:

*"On overall, this study aims at:*

*(a) Assessing the impact of 3DVar in comparison with the EnKF system to predict small-scale extreme weather events initiated over maritime regions with lack of in-situ observations.*

*(b) Compare the forecast impact from assimilating in-situ conventional observations in comparison to assimilating high spatial and temporal resolution data from remote sensing instruments.*

*(c) Provide a quantitative assessment between the different DA schemes by means of using several statistical verification methods."*

We would also like to underscore the novelty of our work: to our knowledge, no previous studies have directly compared 3DVar and EnKF in high-resolution, convective-scale models applied to extreme events initiated over the sea, nor have they evaluated the specific combinations of Doppler radar and satellite observations that we assimilate. We believe this focused comparison offers a valuable contribution to the DA and NWP communities.

**Minor comments:**

*1) NODA simulation: In Section 5 (Model set-up), the configuration of the NODA run is not mentioned. Is it the same as the one used with 3DVar?*

Yes, NODA indeed uses the same model configuration as the 3DVar. Section 5 is not intended to introduce the different numerical experiments. Since Section 5 focuses on the general model setup rather than individual experiments, we added clarification in Section 6 where the experiments are introduced. The revised text in Section 6 now reads as follows:

*"To quantify the benefits of assimilating different observation types with the 3DVar and EnKF DA schemes, a suite of numerical experiments is designed. First, a reference experiment without any data assimilation (NODA), using the same model configuration employed for the WRF experiments performed using 3DVar, is carried out at the regional scales considered in this study."*

*2) Lines 820-830: Regarding the computation of the background error covariance matrix, there is certainly no strict minimum duration, and if it is appropriately calculated, it can lead to improvements in the forecast. However, since the goal of this study is not only to demonstrate improvement compared to the NODA simulation but also to provide a comparison with a method such as EnKF, it becomes crucial that the covariance matrix is computed in the most robust way possible and based on a sufficiently large statistical sample — ideally at least one month, if not a full season (in operational settings, multiple years are often used to ensure robustness). This is particularly important as the background error covariance is one of the cornerstones of variational assimilation (Stanesic et al., 2019). The fact that it was computed over such a short period remains a limitation of the 3DVar approach and should be acknowledged in the comparison with EnKF in the text.*

*Stanesic A, Horvath K, Keresturi E. Comparison of NMC and Ensemble-Based Climatological Background-Error Covariances in an Operational Limited-Area Data Assimilation System. Atmosphere. 2019; 10(10):570. https://doi.org/10.3390/atmos10100570*

We thank the reviewer for highlighting the importance of a robust background error covariance (BEC) matrix, and agree that longer accumulation periods (e.g., a month or season) generally contribute to more reliable variational analysis. However, the choice in this study to use a BEC matrix computed on a short period is based on the CETEMPS extensive experience and previous results (e.g., Hung et al. 2023; Fitzpatrick et al., 2007; Mazzarella et al., 2020, 2021). Moreover, a further improvement in the calculation of the BEC is obtained by applying the recently developed enhancement, which is the inclusion of the CV7 option in WRFDA. This option is particularly beneficial for improving precipitation estimates and the assimilation of radar reflectivity data (Wang et al., 2013; Li et al., 2016; Shen et al., 2022; Ferrer Hernandez et al., 2022), as CV7 utilizes orthogonal functions instead of a vertical recursive filter, leading a more accurate representation of error correlations and potentially improving the quality of our BEC despite the shorter temporal sampling. We would like to emphasize that at CETEMPS the DA for assimilating both radar data and conventional data is based on a BEC matrix computed operationally on the short term because showed the best results on the forecasts.

We have included this discussion in the revised text and now reads as follows:

"... *In this study, we build the 3DVar* ***B*** *matrix over a two-week period, in line with our operational experience running 3DVar and previous demonstrations of its benefits (Hung et al. 2023; Fitzpatrick et al., 2007; Mazzarella et al., 2020, 2021). To enhance* ***B****'s quality despite this relatively short sampling window, we activate the CV7 option in WRFDA. This option uses empirical orthogonal functions (EOFs) to represent vertical covariances instead of the traditional recursive filter, which has proven particularly beneficial for radar-reflectivity assimilation and subsequent precipitation forecast improvements (Wang et al., 2013; Li et al., 2016; Shen et al., 2022; Ferrer Hernandez et al., 2022). In our configuration, the CV7 control variables (i.e., u, v, temperature, pseudo-relative humidity and surface pressure), are defined in EOF space, ensuring a compact yet accurate representation of error structures. We use the CV7 option to generate the* ***B*** *matrix for both case studies. In addition, the weak penalty constraint (WPEC) option (Li et al., 2015) in WRFDA has also been activated to further improve the balance between the wind and thermodynamic state variables, enforcing the quasi-gradient balance on the analysis field."*

*3) Lines 1132-1141: R3 is located near the edges of the domain used and relies solely on in-situ sensors for assimilation corrections, as it lies outside the radar coverage area. This could be another reason for the result obtained.*

We thank the reviewer for this observation. In the revised text, we have updated the discussion of R3 to include this additional factor:

*"In R3, the results show an unexpected behavior when using the moderate threshold (5 mm·h-1) (Fig. 8c), where NODA outperforms DA simulations during the first few hours. This anomaly could be attributed to three factors: (1) the use of a moderate precipitation threshold, which may not capture significant precipitation differences; (2) minimal precipitation in R3 during the initial forecast hours, since the deep convection system had not yet reached this region; and (3) the location of R3 near the domain edges, where it relies solely on in-situ observations for assimilation corrections, as it falls outside the radar coverage area."*

**Reviewer #3:**

**Minor Comments:**

*I am pleased with the revisions, which fully addressed the points raised and makes the outcomes of the study clear. Now the manuscript is in good shape and I recommend minor textual revisions, with no need for another review round.*

We thank the reviewer for all the comments and suggestions that helped us to significantly improve the quality of this study.

*It could be noted that some predictability was already present in the noDA experiment since ECMWF performed assimilation at a somewhat larger spatial scale?*

We thank the reviewer for this insightful observation. Indeed, the **NODA** experiment benefits from the **large-scale ECMWF analysis** on which all ensemble configurations are initialized. Because **all simulations (NODA, 3DVar, and EnKF)** share the same ECMWF background, any advantage conferred by the ECMWF's large-scale assimilation is **common to all experiments**. Highlighting this point in detail risks diverting focus from our primary comparison of the DA methodologies themselves. Consequently, we have chosen not to expand on this aspect in the manuscript but have instead clarified in the text that all runs use the same ECMWF initial conditions.

The beginning of **Section 6** now reads as:

*"To quantify the benefits of assimilating different observation types with the 3DVar and EnKF DA schemes, a suite of numerical experiments is designed. First, a reference experiment without any data assimilation (NODA), using the same model configuration employed for 3DVar, is performed at the regional scales considered in this study. Building on this, several numerical experiments, each differing only in the type of observations assimilated to isolate and compare their impacts on forecast skill, are performed."*

*L143: Wording: "solve" should probably mean "resolve"?*

Done.

*L148: Rephrase "In this context, which DA method is more suitable?". For example: "Given limited computational resources, it is unclear which DA method is more accurate." However, as you know, the answer might depend on how big the resources are.*

Done. We have rephrased the previous sentence as follows:

*"Determining which DA method yields greater accuracy – 3DVar using an ad hoc background error covariance matrix versus EnKF with a flow-dependent low-rank background error covariance derived from a finite ensemble – remains challenging under constrained computational resources."*

*L427-428: "The assimilation of each observation results in a reduction of the ensemble spread, attributed to using a reduced-moderate ensemble size" Confusing. Assimilating an observation reduces the analysis variance in variational and Kalman filter assimilation methods. If you want to motivate adaptive inflation, you could say that the small/finite ensemble sizes shrink the ensemble spread more than it should.*

Agree. We have changed it to the following:

*"Assimilating observations inherently reduces analysis variance in both variational and Kalman filter frameworks. Small ensemble sizes tend to overly collapse the ensemble spread (Anderson and Anderson, 1999). To mitigate this underdispersion and maintain realistic ensemble variance, a spatially varying adaptive inflation technique (Anderson and Collins, 2007; Anderson et al., 2009) is applied to the prior ensemble before assimilating the observations."*

*L429ff: Which technique was applied: spatially varying or homogeneous? According to the citation it was spatially varying?*

Right. The spatially varying technique was applied. We have added this clarification in the manuscript:

*"To mitigate this underdispersion and maintain realistic ensemble variance, a spatially varying adaptive inflation technique (Anderson and Collins, 2007; Anderson et al., 2009) is applied to the prior ensemble before assimilating the observations."*

*Fig 8(i): Data for 3DVar CNTRL is missing. Lower panels: RMSE unit is mm/h?*

We thank the reviewer for noticing this missing. We have added the missing curve for 3DVar CNTRL and added the RMSE units.

[Figure]

*Fig 13: Caption: "Probability of cyclone center occurrence" add for example "(within 20 km)" Check if the values on the colorbars are correct: are all values below 1% or is 0.16 actually 16%? Tick values should appear once on the colorbar. I guess there is a rounding in place.*

We thank the reviewer for these careful checks.

1. **Caption:** We have updated the figure caption to read **"Probability of cyclone center occurrence (within 20 km)"**.

2. **Colorbar values:** The tick labels on the colorbars have been corrected to display each unique value only once after removing unintended rounding.

These changes ensure the figure accurately conveys the intended information.

[Figure]

**Figure 13**. *Probability of cyclone center occurrence (within 20 km) computed using Gaussian KDE for (a) NODA, (b) EnKF (SYN), (c) 3DVar (SYN), (d) EnKF (CNTRL) and (e) 3DVar (CNTRL), from 11 UTC 7 November to 12 UTC 8 November 2014. Qendresa's trajectory observed via satellite imagery is depicted in black.*

*L980-981: Ensemble members are deterministic forecasts, right? If so, replace "deterministic numerical weather models" by "the NODA forecast".*

Done.

*L1060: "much less" ... can you quantify that approximately?*

We thank the reviewer for this request for clarification. Rather than leave the comparison qualitative, we have replaced "much less" with an approximate estimate based on the **ensemble size used in our EnKF experiments**. The main text has modified as follows:

*"Although the EnKF technique has shown in general better performance against the 3DVar for the two extreme weather events analyzed in this study, it is also important to account for the computational resources required by each method. The EnKF requires approximately 36 times more model integrations per cycle than 3DVar's single forecast, in addition to the overhead of computing ensemble updates. This makes the 3DVar appealing because it is much faster and cheaper than the EnKF, and it makes this technique particularly suitable for operational purposes at the small weather forecast centers."*

*L1062-1034: "it does not need either to simulate model trajectories between the assimilation of a set of observations at time t1 and the subsequent set of observations valid at t2" Confusing. Do you mean that 3D-Var simulates one model trajectory, while the EnKF needed to simulate 36 trajectories?*

We thank the reviewer for pointing out the confusion. However, in our previous answer we already decided to remove the entire sentence, so this issue no longer arises in the new version of the manuscript.

*L900-903: "very different" feels vague. Better to be specific. "some members could completely fail in the prediction of the weather event": Does it mean that some members did not predict the existence of a cyclone but just unorganized convection?*

We thank the reviewer for this suggestion. To provide greater clarity, we have replaced the vague phrasing with specific descriptions of member behavior. In the revised manuscript, the text now reads:

*"The low predictability of Qendresa and the high sensitivity to physical parameterizations produce substantial spread in ensemble behavior: some members capture the cyclone's closed circulation and track reasonably well, while others fail to develop a coherent low-pressure core, instead producing only disorganized or weak convective cells. Consequently, these poorly performing members may entirely miss the medicane's formation or misplace its center, leading to large errors in both track and intensity forecasts."*

*L903-904: " In this situation, our small-to-moderate ensemble will probably produce a poor flowdependent background error covariance matrix": If it is somewhat probable that there is no cyclone, then this information should be in the background error covariance, I would say. However, you might mean that large uncertainty/spread leads to substantial nonlinearity, which is detrimental to the analysis accuracy of the EnKF.*

We thank the reviewer for this insightful comment. You are correct that, in principle, a perfectly represented background error covariance matrix would reflect all uncertainties, including the possibility of no cyclone. In practice, however, finite‑sized ensembles suffer from sampling error, which leads to spurious correlations and underdispersion, issues that are exacerbated when the underlying model forecasts fail to represent the event accurately.

To clarify, we have revised the text to read:

*"In this situation, our small‑to‑moderate ensemble size exacerbates sampling error, yielding spurious background error covariances that degrade analysis accuracy in the EnKF. These errors become particularly problematic when the numerical model mispredicts the event, since the ensemble members no longer provide a reliable representation of flow‑dependent uncertainty."*

*L905-907: " for which the ensemble mean will be smoothed out significantly". The ensemble mean should not be expected to be a good forecast of the true state, in case the distribution is non-Gaussian, which it will be for extreme precipitation and the cyclone's pressure field.*

Agree. Since this sentence has been removed in the revised manuscript, we confirm that no further action is needed on this point.

*L909-911: it is important to note that although the ensemble mean of the EnKF_SYN is not correctly reproducing the intensification of Qendresa, some of the ensemble members very well reproduce the*

*observed MSLP both in deepening and timing" In the light of the above comment, it should not be unexpected that averaging removes extreme values.*

Agree.

*L1069: "the 3DVar performs better than the EnKF ensemble mean" Yes, but then again, this is likely for extreme events because the ensemble mean is averaging over the skewed probability density function. I suggest rephrasing, since obviously in probabilistic metrics, like ROC/AUC, the situation is reversed.*

We appreciate the reviewer for pointing out this. We have modified the main text as follows:

*"An interesting result of this study is that, for highly non‑Gaussian extreme events the deterministic 3DVar forecast can occasionally outperform the EnKF ensemble mean in terms of point forecasts (e.g., minimum central pressure), because averaging across ensemble members tends to smooth out the tails of a skewed probability distribution. In contrast, probabilistic metrics like ROC/AUC consistently favor the EnKF, reflecting its superior ability to capture forecast uncertainty. We attribute these contrasting behaviors to the different approaches to background error covariances: 3DVar employs a static covariance, while EnKF uses a flow‑dependent covariance estimated from a finite ensemble. To combine the strengths of both methods, a hybrid error covariance approach—where the forecast error covariance matrix is formed by linearly blending the EnKF's ensemble‑derived covariances with the 3DVar's static climatological covariances—may offer improved forecast skill for convective‑scale extreme events."*